# Accurate nowcasting of cloud cover at solar photovoltaic plants using geostationary satellite images

Pan Xia[1], Lu Zhang[2], Min Min [1] ✉, Jun Li[2], Yun Wang[3], Yu Yu[4] & Shengjie Jia[5]

Accurate nowcasting for cloud fraction is still intractable challenge for stable solar photovoltaic electricity generation. By combining continuous radiance images measured by geostationary satellite and an advanced recurrent neural network, we develop a nowcasting algorithm for predicting cloud fraction at the leading time of 0–4 h at photovoltaic plants. Based on this algorithm, a cyclically updated prediction system is also established and tested at five photovoltaic plants and several stations with cloud fraction observations in China. The results demonstrate that the cloud fraction nowcasting is efficient, high quality and adaptable. Particularly, it shows an excellent forecast performance within the first 2-hour leading time, with an average correlation coefficient close to 0.8 between the predicted clear sky ratio and actual power generation at photovoltaic plants. Our findings highlight the benefits and potential of this technique to improve the competitiveness of solar photovoltaic energy in electricity market.

Reducing fossil fuel use and global climate change requires a fast energy transition, and nations across the globe have successively set out their own targets and pathways to carbon neutrality[1]. Since 2009, as the fastest-growing renewable power source, the generating capacity of solar photovoltaic (PV) energy has grown globally by 41% per year[2]. It has put forward higher requirements for the conversion efficiency and capital cost reduction of PV energy generation[3], which is always impacted by cloud cover, aerosol and panel soiling[4–9]. Yet, in a stark contrast to aerosol and panel soiling, cloud cover or advection can dramatically and intermittently affect incident solar radiation, resulting in unbalance between the load demand and PV energy generation, which poses a considerable risk to the stability of power grids[10–12]. Therefore, reliable and powerful PV energy generation or global tilted irradiance (GTI, the radiation captured by solar photovoltaic panels) forecast technique, particularly short-term forecasts of the intra-day GTI or PV power generation (at the leading time of 0–4 h),

is also highly beneficial to power smoothing processes and other load-following applications[9,13]. In addition, currently, in most European countries, short-term prices for the physical delivery of electricity are formed by spot markets, such as the European Power Exchange SPOT (https://www.europex.org/members/epex-spot/). Although ~80% of trade volume is controlled by the day-ahead trading market, the intra-day auctions from hourly to 15-min intervals determine real-time electricity prices[14]. Thus, sophisticated solar PV power generation nowcasting technique not only can improve the stability of power generation, but also facilitates the developments of more commercially viable PV systems, the current electricity market and price transactions, and increases the competitiveness of the solar PV energy source[15,16].

In recent years, rapid advances in artificial intelligence have promoted the application of data-driven machine learning-based approaches in Earth system science[17,18]. Particularly, some recent

[1]School of Atmospheric Sciences and Guangdong Province Key Laboratory for Climate Change and Natural Disaster Studies, Sun Yat-sen University and Southern Marine Science and Engineering Guangdong Laboratory (Zhuhai), Zhuhai 519082, China. [2]Key Laboratory of Radiometric Calibration and Validation for Environmental Satellites and Innovation Center for FengYun Meteorological Satellite (FYSIC), National Satellite Meteorological Center (National Center for Space Weather), China Meteorological Administration, Beijing 100081, China. [3]China General Nuclear Power Group (CGN) Wind Energy Co Ltd, Beijing 100106, China. [4]National Meteorological Information Centre, China Meteorological Administration, Beijing 100081, China. [5]Beijing Keytec Technology Co., Ltd., Beijing 100081, China. ✉e-mail: minm5@mail.sysu.edu.cn

studies[12,13,19–21] on the prediction of solar radiation also explicitly indicate that advanced prediction approaches based on machine learning perform better compared with empirical models, time series, and hybrid algorithms, such as artificial neural networks and support vector machines. Nevertheless, it is still a great challenge to predict cloud motion, formation, deformation and dissipation under complex atmospheric dynamics, geography, and climatic conditions[9,22,23]. Thus, there is still no solar radiation forecast model that can work well in every region and at every time[21].

Cloud cover nowcasting remains a field of interest for forecasting the electricity production of PV plants[24]. We are committed to developing a daytime hourly intra-day cloud fraction (CF) prediction algorithm for small areas over PV plants. Based on the recurrent-neural-networks-based (RNNs) long short-term memory (LSTM) algorithm framework, the newly developed PredRNN and PredRNN++ (an extended and latest version of PredRNN)[25,26] can well learn to predict long-term future imageries in various spatio-temporal tasks by modeling their spatial and temporal dependencies, including video frame prediction, human motion prediction, etc. Therefore, our primary objective is to develop an innovative and easy-to-promote algorithm or system based on the key framework of the PredRNN++ model. Through this algorithm, the 0–4 h CF at solar PV plants under all weather conditions can be predicted by using sequential Himawari-8/9 geostationary satellite images with high spatio-temporal resolutions[27]. Compared with the previous study[28], it only used a single visible channel of geostationary satellite and a constant model to predict cloudiness. Some former studies directly used surface solar global horizontal irradiance (GHI) as model input to predict GHI values in the next few hours[29], achieving the purpose of estimating the power generation of PV plants. Nevertheless, the presence of clouds is still identified as the primary uncertainty in current surface solar GHI forecasts[30]. In contrast, our investigation only predicts geostationary satellite Level 1B (L1B) radiance data. With the prediction results of

satellite L1B radiance data and accurate cloud detection algorithm, this approach is expected to provide reliable and variable CF information for further improving the predictability of current GTI or PV power generation.

## Results

### Cloud fraction nowcasting and validations

In order to better simulate real application scenarios, a quasi-operational and near real-time (NRT) and cyclically updated prediction system is newly developed for 0–4 h CF nowcasting at solar PV plants (hereafter referred to as the NCP_CF). The predicted CF from this NCP_CF nowcasting system is also compared with the real PV power generation and the GTI to verify its feasibility, reliability and adaptability. The results from the NCP_CF system are examined and validated by using the observed CF values from twelve manual meteorological observation stations of the China Meteorological Administration (only the observations at 14:00 and 17:00 are used) and three all-sky imager stations (Fig. 1). Figure 2 shows the root mean square errors (RMSEs) and mean bias errors (MBEs) of the CF predictions at targeted stations with the forecast horizon, local time (diurnal cycle) and time series. In terms of forecast horizon (Figs. 2a–2f, Supplementary Table 1), the RMSE increases from 0.18 to 0.35 at all stations for 0–4 h forecast periods, and the MBE fluctuates around −0.1. Notably, the RMSE is less than 0.25 within the 2-h leading period, but the forecast accuracy decreases faster when the forecast leading period exceeds 2 h, indicating that the forecast performance threshold of this system is ~2-h leading time. Considering the continuity and coverage of observation time, Fig. 2g only shows the diurnal cycle of CF forecast accuracy at three all-sky imager stations. Within a one-day forecast window, the most and least accurate predictions occur during 12:00–17:00 and 08:00–09:30, respectively, whereas the relatively moderate decrease in the forecast accuracy before 09:30 is mainly attributed to the invalid satellite visible images before 08:00.

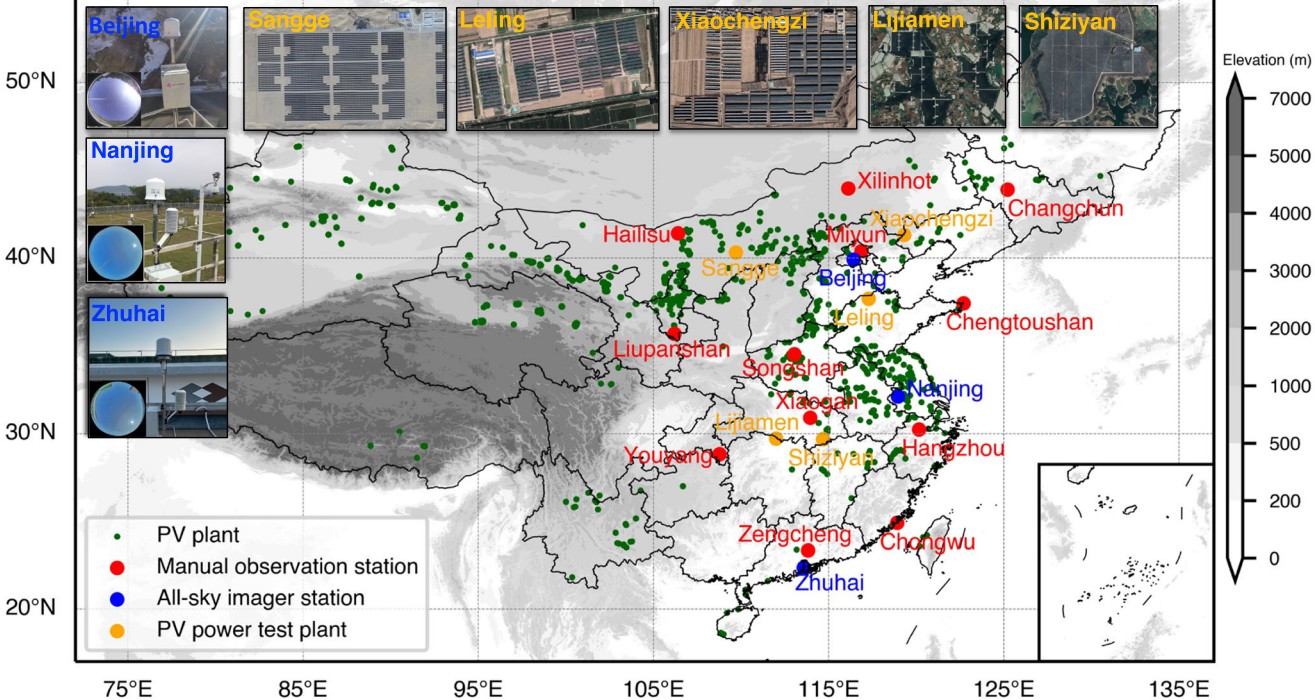

**Fig. 1 | Geographical distributions and photographs of ground-based sites.** PV plants (dark green small solid circles), 12 manual observation stations (red solid circles), 3 all-sky imager stations (blue solid circles), and 5 PV power test plants (yellow solid circles). The photos above the PV plants are exported from Google Earth Pro. Sangge Map Data: Google, mage ©2023 CNES/Airbus; Leling Map Data: Google, Image ©2023 Maxar Technologies; Xiaochengzi Map Data: Google, Image ©2023 Maxar Technologies; Lijiamen Map Data: Google, Image ©2023 CNES/Airbus; Shiziyan Map Data: Google, Image ©2023 CNES/Airbus. Source data are provided as a Source Data file.

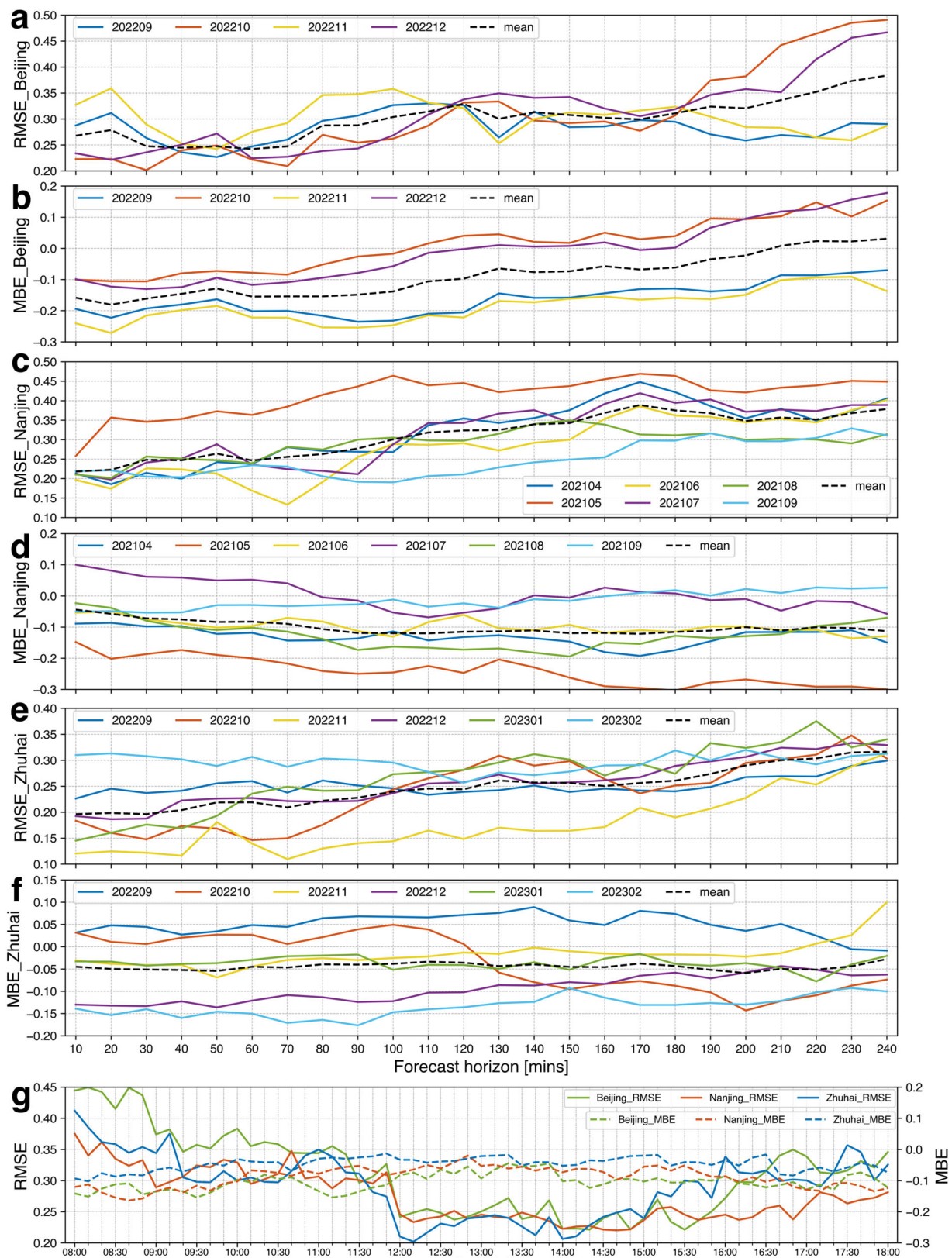

**Fig. 2 | RMSE and MBE between the predicted CF and the CF obtained by all-sky imagers.** Subfigures (**a**), (**c**), (**e**) and (**b**), (**d**), (**f**) are the RMSEs and MBEs for Beijing, Nanjing and Zhuhai all-sky imager stations, respectively. The different colored lines represent the results for different months, and the dashed black line represents the mean of all the lines. Subfigure (**g**) depicts the mean RMSE and MBE of three all-sky imager sites at different local time of one day. Source data are provided as a Source Data file.

Regarding the predicted performance for different months, the monthly mean RMSE and MBE values in time series at three all-sky imager stations do not vary considerably, indicating the weak monthly dependence and stability of this cyclically updated CF nowcasting algorithm and system.

Furthermore, the correlation coefficients (R) between the 1–4 h predicted clear sky ratios (CSRs; CSR = 1−CF) from the NCP_CF system and the actual power generation (GTI) are calculated. Figure 3 displays the comparisons among the 1–4 h predicted CSRs, the actual power generation and the GTI at Sangge, Leling, Xiaochengzi, Lijiamen, and Shiziyan PV plants from 09:00 to 17:00 in November 2022. The mean R values between the 1–4 h predicted CSRs and the actual PV power generation (the GTI) in November 2022 at five PV plants are 0.81, 0.73, 0.65, and 0.55 (0.81, 0.72, 0.64, and 0.55). This result highlights the good consistency of the predicted CSRs with the actual PV power generation and the GTI, especially for the first 2-h leading time. Overall, the EPM-model-based NCP_CF system developed in this research is applicable to provide high-quality CF estimations at PV plants in advance, which can be used to predict the GTI and power generation at the forecast leading time of 0–4 h. The results at Sangge and Leling PV plants from December 2022 to March 2023 are displayed in Supplementary Figs. 3–6.

### Near real-time and cyclically updated prediction system

The satellite-based NRT and cyclically updated prediction system (Fig. 4) for 0–4 h CF nowcasting, operating at five real PV plants (Sangge, Leling, Xiaochengzi, Lijiamen and Shiziyan PV plants) belonged to China General Nuclear Power Group Wind Energy Co. Ltd., mainly consists of three subsystems, i.e., preprocessing, prediction and retrieval modules. Specifically, the preprocessing module regularly adjusts the real-time down-sampling Himawari-8/9 Advanced Himawari Imager (AHI) data received from the direct broadcast receiving system. The AHI is an advanced imager with 16 spectral bands ranging from 0.47 μm to 13.3 μm, which has spatio-temporal resolutions of 4 km and 10 min[31]. The full-disk Level-1B radiance data at 0.65 μm, 0.86 μm, 3.9 μm, 7.0 μm, 11.2 μm and 12.3 μm with high-quality geolocation and radiometric calibration is grided into a $32 \times 32$ pixel box (-128 km × 128 km) centered around the targeted PV plant. Then, these sequential and resized AHI images at six bands are converted into a tensor [tile size = 6 (band number) × 24 (4-h time sequence) × 32 × 32] of the prediction model for forecasting the following 0–4 h satellite images ($6 \times 24 \times 32 \times 32$).

As a key function of the prediction module, an enhanced PredRNN + + model (EPM; more details in "Model" section) is developed in this study to predict 0–4 h sequential geostationary satellite images. In order to better track the fast and stochastic changes in cloud images, the neural network of the EPM is always cyclically generated by the scheduled training process of the model, which has a 1-h update frequency. The cumbersome cyclic process should take 40–50 min when using the latest sequential satellite images ($6 \times 24 \times 32 \times 32$) between −5 h and −1 h as training samples and one graphics processing unit processor (NVIDIA Tesla-V100). For instance, a cyclic training process of the model starts at the scheduled local time of 12:08 and ends at ~12:53, the imported training samples are from 08:00 to 12:00, and the latest updated EPM is timely activated for CF nowcasting at ~13:08 (nowcasting from 13:00 to 17:00) and 13:38 (nowcasting from 13:30 to 17:30), respectively, with an update frequency of a frequency of half an hour. During the same period, the network of the next EPM is also trained simultaneously by using the sequential satellite images from 09:00 to 13:00. This cyclical procedure will continually update and replace the existing EPM every hour, ensuring that we always have the most up-to-date nowcasting model. Considering the use of satellite visible images, the NCP_CF system only operates from 07:20 to 17:20, which still meets the requirement of CF nowcasting at PV plants.

In the retrieval module, the 0–4 h predicted and resized cloud images mentioned above are used by a fast cloud mask algorithm[32], which is able to calculate the number of cloudy pixels and the CF or cloud cover in the observation field of the targeted PV plant. Note that compared with the operational cloud mask algorithm, the fast cloud mask algorithm can fast retrieve the CF without any ancillary data, which is crucial for CF nowcasting. Supplementary Figure 1 presents the comparisons between the predicted and actual satellite images and cloud mask results at Zhuhai station on 17 November 2022, illustrating the good agreement between them.

## Discussion

Our study demonstrates that the NCP_CF system can provide high-efficiency, high-quality and adaptable 0–4 h CF nowcasting data for PV plants. As shown in Figs. 2a–f, the mean RMSE (MBE) values are 0.21 (−0.09), 0.25 (−0.08), 0.3 (−0.07), and 0.35 (−0.03) for the forecast leading time of 1 h, 2 h, 3 h, and 4 h, respectively. Particularly, the CF nowcasting results from the NCP_CF system remain highly reliable within the forecast leading time of 2 h, with RMSE values staying almost at 0.2 and not increasing within the 1-h leading time. Conversely, the prediction performance of the NCP_CF system gradually deteriorates as the forecast leading time increases to more than 2 h, which may be due to the vanishing gradient problem[22]. By the limited spatial domain, the rapid movement of clouds may cause a small bias between the predicted CF and the actual CF. Further analyses on the daily and seasonal scales are also conducted, as shown in Fig. 2g, a–f, respectively. On the daily scale, the NCP_CF system performs particularly well from 09:30 to 18:00, whereas it shows poor performance before dawn, mainly due to the poor quality of satellite data at the visible band during that time. Fortunately, this issue is mitigated due to the low power generation of PV plants before dawn, and thus cloud cover has a low impact on power generation in this period. For the seasonal scale, except for the forecast results at Nanjing station in May 2021, the NCP_CF system shows stable forecast performance and seasonal biases at different stations and in different seasons. Its salient adaptability thus is the largest advantage compared with other solar radiation nowcasting methods summarized in the previous review[21]. Overall, the CF nowcasting results of the NCP_CF system have good stability, strong generalizability and non-sensitivity to geographical locations and climatic characteristics.

Given that the present electricity spot markets in Europe work within different time horizons, specific and professional forecast techniques are required for each leading time[13]. The NCP_CF system with the 0–4 h forecast leading time within a 10-min interval shows more advantages than other existing nowcasting methods, such as the manners based on all-sky imager observation (the forecast leading time only ranging from 0 to 20 mins)[33] and numerical weather prediction (from 6 h to day-ahead time frames)[34] for solar PV power generation. Principally, this system is applicable to fast varying small-scale weather and environmental conditions and can accurately capture cloud motion over PV plants without relying on long-term historical in-situ meteorological data. Besides, the predictions of the NCP_CF system are not markedly affected by seasonal climate changes on long-term scales, which underscores the stable operation of this system. The system shows excellent forecast performance within the first 2-h leading time, with an average R value between the predicted CF and the actual power generation or GTI at PV plants close to or more than 0.80.

Since one of the greatest challenges facing solar PV renewable energy production is its instability and intermittency, accurate CF nowcasting is still vital for the efficient operation of PV plants and their power systems. Improving the stability of PV power production can directly facilitate policy-making of feed-in tariffs and attract more investment in solar PV power generation[14,35]. However, most importantly, the nowcasting technique developed in this research deserves

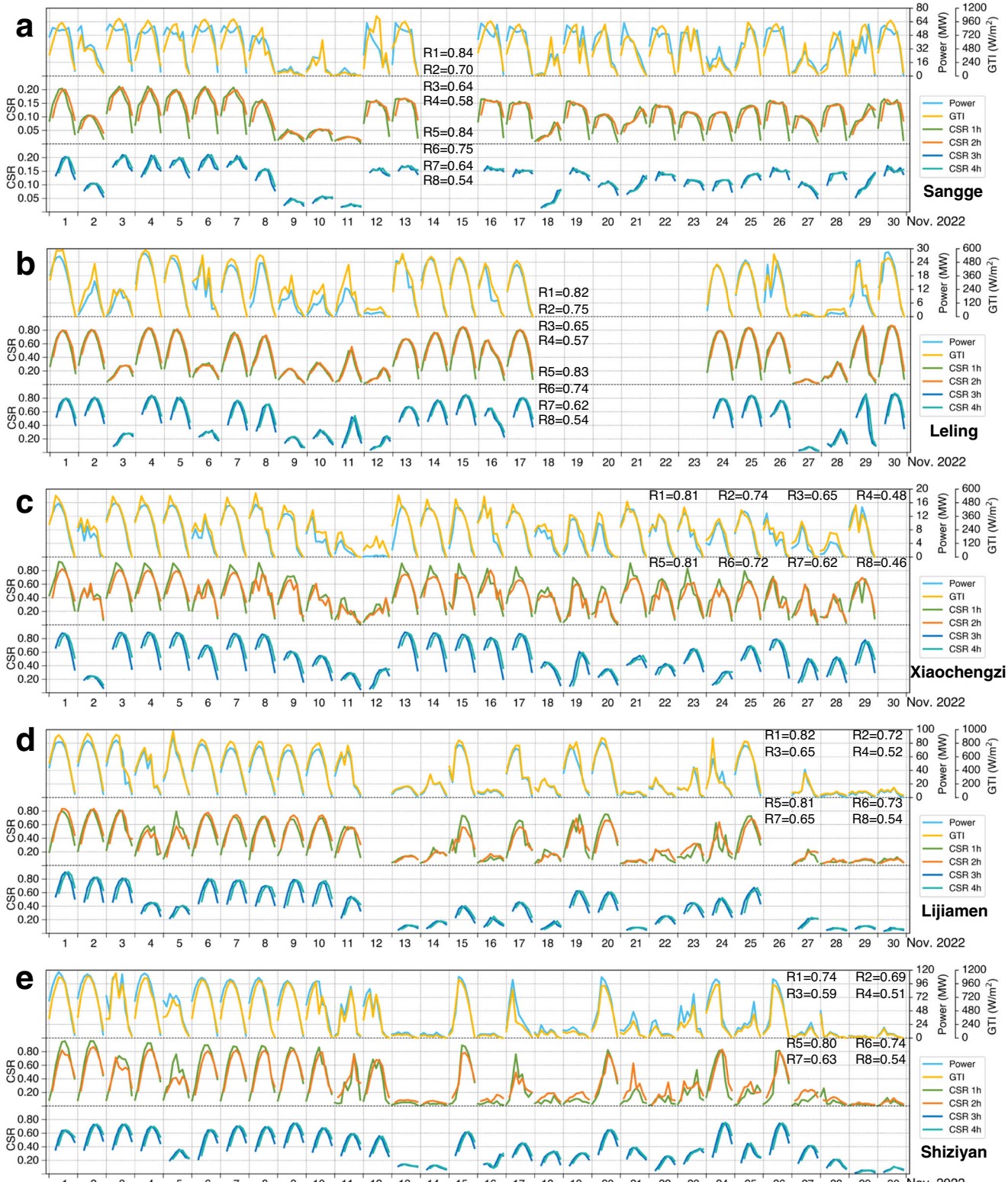

**Fig. 3 | Power and GTI of PV plants and corresponding 0–4 h CSR.** Time series of the power (MW), GTI (W·m⁻²) and predicted clear sky ratio (CSR) at (**a**) Sangge, (**b**) Leling, (**c**) Xiaochengzi, (**d**) Lijiamen, and (**e**) Shiziyan PV plants from 09:00 to 17:00 (local time or Beijing time) on each day in November of 2022. R1–R4 and R5–R8 indicate the correlation coefficients (R) of the predicted CSR with the power and the GTI for forecast leading time of 1–4 h, respectively. Note that the missing and invalid data are not shown. Source data are provided as a Source Data file.

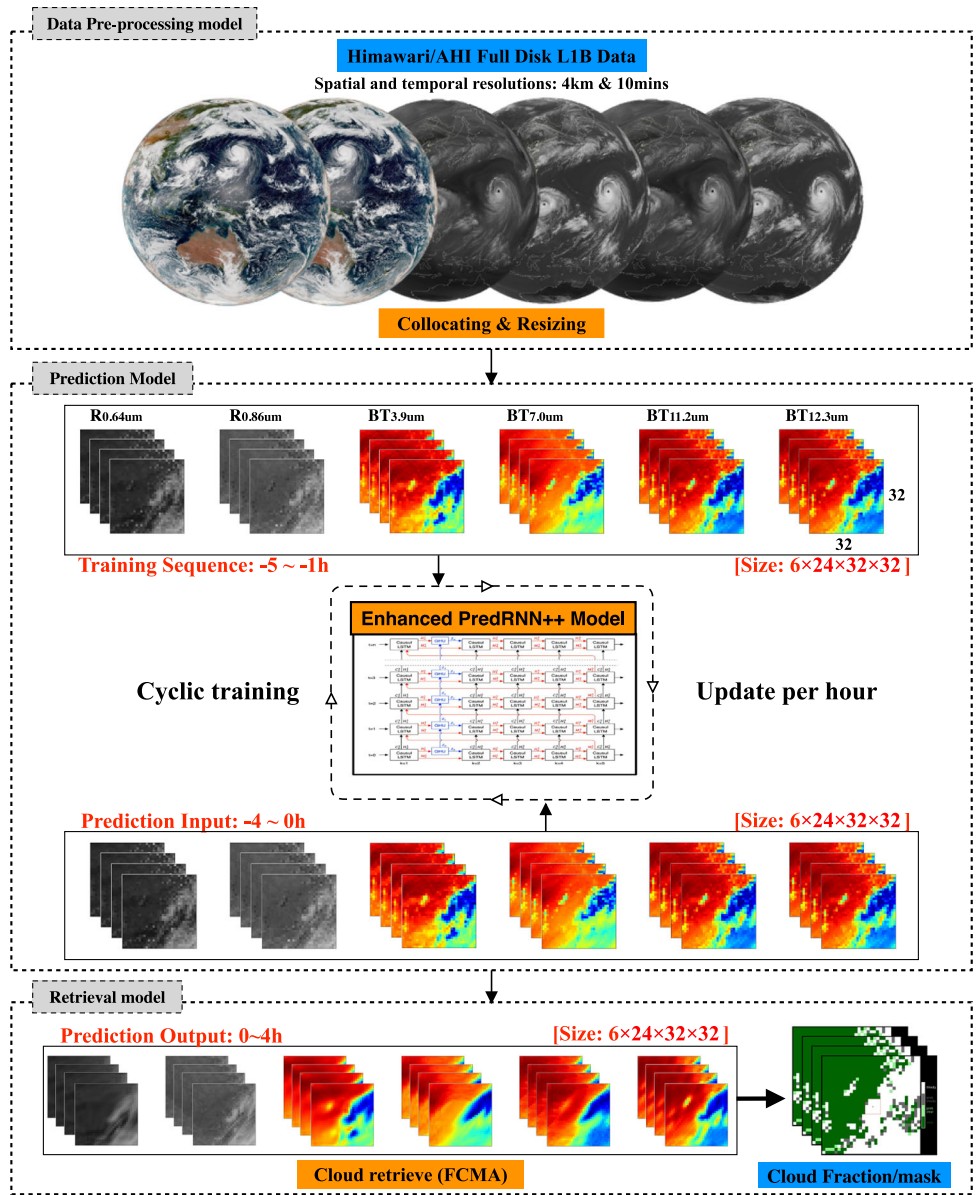

**Fig. 4 | CF prediction system schematic diagram.** The near real-time and dynamically updated prediction system for the cloud image and fraction nowcasting at the leading time of 0–4 h at the photovoltaic (PV) plants. For satellite images in this figure, red (white) color signifies high temperature (reflectance) and blue (black) color represents low temperature (reflectance). About Cloud Fraction pictures, The white, gray, light green, and dark green spots represent cloudy, probably cloudy, probably clear and clear pixel labels, respectively.

attention in terms of promoting the overall penetration of solar power on the electric grid and having a non-negligible impact on electricity price trading in the intra-day spot market[14]. Furthermore, it is evident that increasing the share of renewable energy in the global energy system can contribute to the reduction of global carbon emissions[3]. Therefore, our future mission is to further promote applications and improve the accuracy of this cloud cover nowcasting technique, especially for the forecast leading time of more than 2 h, by using higher spatial-resolution satellite data (i.e., 1–2 km) and combining the short-term forecast data from a rapidly updated regional high-resolution numerical weather prediction.

## Methods
The newly developed EPM and fast cloud mask algorithm in the NCP_CF system are applied to predict CF (or CSR) at the leading time of 0–4 h at two test PV plants. In this system, the −4–0 h geostationary satellite radiance data is used as input to dynamically provide 0–4 h

satellite cloud images and fractions. To verify the reliability of the EPM, we first use the CF observations from twelve widely distributed ground-based manual stations and three all-sky imager stations for the period from 2019 to 2022 as true values to compare with the predictions in the corresponding period. Moreover, correlations of the predicted CSRs with the actual PV power generation and surface solar radiation at five test PV plants from October 2022 to March 2023 are analyzed. The benefits of the geostationary satellite data with high spatio-temporal resolutions and the advanced EPM to improve the PV power generation efficiency are investigated, as well as the wide applicability and generalizable value of the EPM system.

### Ground-based observation data
The total cloud cover or CF (~20 km × 20 km square area) used in this study is obtained through manual observation at twelve ground-based meteorological stations (Fig. 1) in January, April, July, and October of 2019. Note that due to the relatively large errors in low-visibility

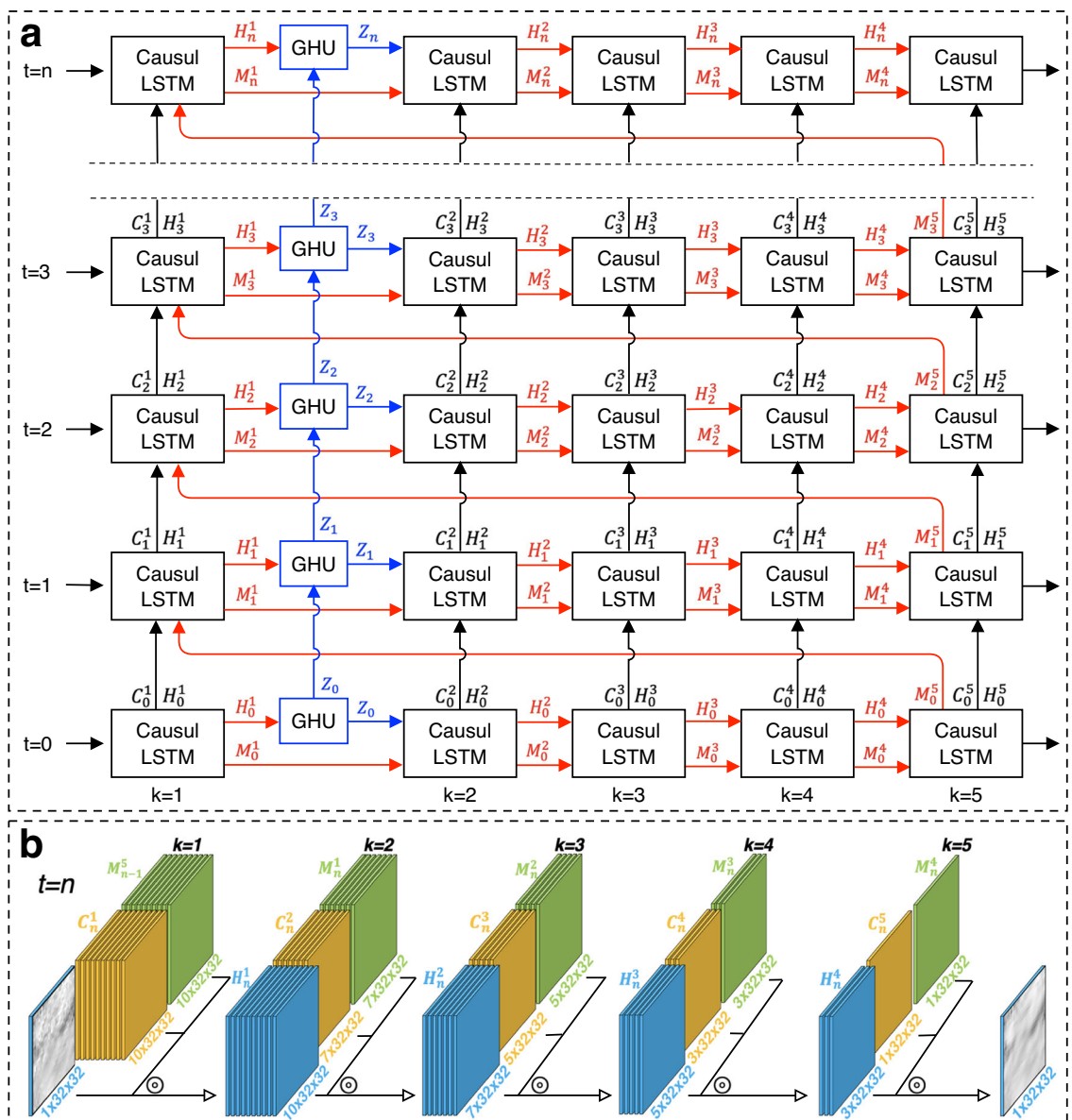

**Fig. 5 | Enhanced PredRNN++. a** Framework of the enhanced PredRNN++ model with five convolutional layers, and (**b**) visual illustration of the flow of input data in the spatial memory. In Fig. 5a, the Gradient Highway Unit (GHU; blue) is embedded between the first and the second convolutional layers, the horizontal red arrows denote the deep transition paths of the spatial memory, the vertical black arrows represent the updating direction of the temporal memory, and the blue parts indicate the gradient highway connecting the current time step directly with previous inputs. In Fig. 5b, "⊙" denotes the convolution, $C_n^k$ the temporal memory, $M_n^k$ the spatial memory, and $H_n^k$ the hidden state. $M_n^{k-1}$, $C_n^k$ and $H_n^{k-1}$ are concatenated to form a larger tensor, and then $H_n^k$ is generated by the convolution. The numbers below each memory indicate the dimensions of the corresponding tensors.

conditions, the CF data with the matched and automatically measured visibility <2 km is removed. Besides, the view zenith angles of ground-based stations from H8/AHI field of view used here, as stated in Supplementary Table 2, are smaller than 60°. Therefore, the parallax effect is negligible (error < 1 km) in the collocation between satellite pixels and ground-based stations for this study (For more explanations and details, please refer to Supplementary Note and Supplementary Figs. 7 and 8). Firstly, considering the daytime nowcasting applications and sunshine conditions at PV plants, only the manual observations at 11:00, 14:00 and 17:00 are collected for validation in this research. Secondly, three ground-based all-sky imager stations (equipped with a Japan EKO ASI-16 all-sky imager, https://www.eko-instruments.com/us/categories/products/all-sky-imagers/asi-16-all-sky-imager) can provide the high temporal resolution (5 min) and valuable CF and cloud cover data during the daytime, which are retrieved by the standard EKO ASI-

16 cloud detection algorithm (https://www.eko-instruments.com/media/z2aalysq/asi-16-software-manual-find-clouds.pdf)[36]. ASI, equipped with a digital camera coupled with an upward looking fisheye lens, could provide field of view (FOV) of ~180°, but pixels at a FOV > 140 ° are excluded due to distortion. Digital images of the sky obtained by ASI are classified pixel by pixel into clear sky, optically thin and optically thick clouds, respectively. The cloud detection and opacity classification (CDOC) algorithm developed by Ghonima et al., 2012[36] could provide 96%, 60%, and 96.3% accuracy in the validation for clear, thin, and thick cloud, respectively. Finer CF data allow more accurate validation of the NCP_CF system, thereby demonstrating the specific prediction effect at the forecast leading time of 0−4 h. The study periods of the local CF data from three all-sky imagers located in Zhuhai, Nanjing and Beijing (Fig. 1) are from September 2022 to February 2023, April 2021 to September 2021, and September 2022 to

December 2022, respectively. In addition, the actual power generation (MW, temporal resolution of 15 min) and GTI (W·m$^{-2}$) measured at Sangge and Leling PV plants from November 2022 to March 2023 and at Xiaochengzi, Lijiamen and Shiziyan PV plants in November 2022 (Fig. 1) are also used to analyze the agreement with the predicted CF. These real power generation data are obtained from the SCADA (Supervisory Control and Data Acquisition) system of China General Nuclear Power Group Wind Energy Co. Ltd.

## Geostationary satellite data and calculating the cloud fraction

The NRT 16-band full-disk AHI level-1B radiance data from the Himawari-8/9 satellite (the new-generation Japanese geostationary meteorological satellite) with spatio-temporal resolutions of 1–4 km and 10 min are obtained from the Japan Meteorological Agency Himawari-Cast in China[32]. Additionally, the offline Himawari-8/9 data at the original resolution (0.5–2 km) are also available for free download from the JAXA (Japan Aerospace Exploration Agency) Himawari satellite data FTP (File Transfer Protocol) site (ftp.ptree.-jaxa.jp) from July 7, 2015 (http://www.jma-net.go.jp/msc/en/). The nadir point of the Himawari-8/9 satellite is located at 140.7°E, and the coverage of this satellite includes the Japan island and the eastern parts of China.

Based on the real-time Himawari-8/9 AHI full-disk observation data, each site would be centered around its precise location and matched with a 32 × 32 pixels box as the experimental area. The special fast cloud mask algorithm[32] combines five inherited and improved cloudy/clear pixel tests in visible and infrared bands to determine the final confidence value ($c$) of every pixel of the satellite imager, i.e., $c > 0.99$ = clear, $0.95 < c \leq 0.99$ = probably clear, $0.66 < c \leq 0.95$ = probably cloudy, and $c \leq 0.66$ = cloudy. As the real viewing field at a ground-based station approximates a 20 km × 20 km square area, a 5 × 5 pixels box of cloud mask centered around a targeted PV plant is used in this study to calculate the CF predictions, which is expressed as Eq. (1).

$$\text{CF} = (\text{Num}_{cloudy} + \text{Num}_{prob-cloudy})/25 \qquad (1)$$

where $\text{Num}_{cloudy}$ and $\text{Num}_{prob-cloudy}$ indicate the total numbers of the cloudy and probably cloudy pixels in the 5 × 5 pixel box[37], respectively. The complementary CSR is equal to 1-CF.

## Model

The PredRNN++ model[24] (an improved prediction RNN), dedicated to short-term prediction and nowcasting, is adopted as a key model in this investigation for 0–4 h CF nowcasting. This advanced neural network successfully overcomes the spatio-temporal predictive learning dilemma between deep-in-time structure and vanishing gradient. Previous research has demonstrated that the PredRNN++ consistently outperforms the ConvLSTM, TrajGRU, Discrete Fracture Network, MCnet and PredRNN at every future time step for both peak signal-to-noise ratio and structural similarity index measure[24]. To achieve greater spatio-temporal modeling capability, in this investigation, we re-design and develop the EPM with five convolutional layers, whose elaborated structure is shown in Fig. 5. The details of the casual LSTM and the Gradient Highway Unit (GHU)[24] in the EPM structure are also illustrated in Supplementary Fig. 2.

The causal LSTM, an upgraded version of the LSTM, increases the recurrence depth from one time step to the next and derives a more powerful modeling capability for stronger spatial correlations and short-term dynamics. As shown in Supplementary Fig. 2a, a causal LSTM unit contains two memories, namely a temporal memory $C_t^k$ and a spatial memory $M_t^k$, where the superscripts $k$ and $t$ denote the $k$th hidden layer in the stacked causal LSTM network and the $t$th time step, respectively. The temporal memory $C_t^k$ depends on its preceding state $C_{t-1}^k$ and is controlled by an input gate $i_t$, a forget gate $f_t$ and an input

modulation gate $g_t$. The spatial memory $M_t^k$ relies on $M_t^{k-1}$ which is in the deep transition route. Notably, the topmost spatial memory $M_{t-1}^5$ is assigned to the bottom spatial memory $M_t^0$. For the $k$th layer, the updated equations of the causal LSTM can be expressed as Eqs. (2–7).

$$\begin{pmatrix} g_t \\ i_t \\ f_t \end{pmatrix} = \begin{pmatrix} \tanh \\ \sigma \\ \sigma \end{pmatrix} W_1 \odot [X_t, H_{t-1}^k, C_{t-1}^k], \qquad (2)$$

$$C_t^k = g_t \otimes i_t + f_t \otimes C_{t-1}^k, \qquad (3)$$

$$\begin{pmatrix} g_t' \\ i_t' \\ f_t' \end{pmatrix} = \begin{pmatrix} \tanh \\ \sigma \\ \sigma \end{pmatrix} W_2 \odot [X_t, C_t^k, M_t^{k-1}], \qquad (4)$$

$$M_t^k = g_t' \otimes i_t' + f_t' \otimes \tanh(W_3 \odot M_t^{k-1}), \qquad (5)$$

$$O_t = \tanh(W_4 \odot [X_t, C_t^k, M_t^k]), \qquad (6)$$

$$H_t^k = O_t \otimes \tanh(W_5 \odot [C_t^k, M_t^k]), \qquad (7)$$

where "$\odot$" denotes the convolution, "$\otimes$" the element-wise multiplication, tan$h$ the element-wise hyperbolic tangent function, σ the element-wise sigmoid function, and "[]" a concatenation of tensors. $W_{1-5}$ the convolutional filters. All the equations in the causal LSTM can be briefly expressed as Eq. (8).

$$H_t^k, C_t^k, M_t^k = CauLSTM_k(H_t^{k-1}, H_{t-1}^k, C_{t-1}^k, M_t^{k-1}), \qquad (8)$$

where $H_t^{k-1}$ is replaced by $X_t$ and $Z_t$ between the first and second layers.

The gradient highway, a shorter route from future outputs back to distant inputs, can alleviate the vanishing gradient problem. As shown in Supplementary Fig. 2b, in a GHU, $W$ represents the convolutional filters, $S_t$ is a switch gate and enables adaptive learning between the transformed input $P_t$ and the hidden state $Z_{t-1}$. The equation of the GHU can be written as Eqs. (9–11).

$$P_t = \tanh(W_{pz} \odot Z_{t-1} + W_{ph} \odot H_t^1), \qquad (9)$$

$$S_t = \sigma(W_{sz} \odot Z_{t-1} + W_{sh} \odot H_t^1), \qquad (10)$$

$$Z_t = S_t \otimes P_t + (1 - S_t) \otimes Z_{t-1} \qquad (11)$$

The equations of the GHU can be briefly expressed as:

$$Z_t = GHU(H_t^1, Z_{t-1}) \qquad (12)$$

As presented in Fig. 5a, combined with Eqs. (8) and (12), the key equations of the entire EPM framework can be written as Eqs. (13–18).

$$H_t^1, C_t^1, M_t^1 = CauLSTM_1(X_t, H_{t-1}^1, C_{t-1}^1, M_{t-1}^5) \qquad (13)$$

$$Z_t = GHU(H_t^1, Z_{t-1}) \qquad (14)$$

$$H_t^2, C_t^2, M_t^2 = CauLSTM_2(Z_t, H_{t-1}^1, C_{t-1}^1, M_t^1), \qquad (15)$$

$$H_t^3, C_t^3, M_t^3 = CauLSTM_3(H_t^2, H_{t-1}^3, C_{t-1}^3, M_t^2), \quad (16)$$

$$H_t^4, C_t^4, M_t^4 = CauLSTM_4(H_t^3, H_{t-1}^4, C_{t-1}^4, M_t^3), \quad (17)$$

$$H_t^5, C_t^5, M_t^5 = CauLSTM_5(H_t^4, H_{t-1}^5, C_{t-1}^5, M_t^4), \quad (18)$$

In the EPM framework, the GHU is injected between the first and second causal LSTMs. The causal LSTM and GUH respectively capture short-term and long-term data or image dependencies. The gradient highway (blue line in Fig. 5a) supplies a quick path from the first to the last time step by quickly updating hidden state $Z_t$. It is worth noting that, unlike temporal skip connections, the GHU controls the proportions of $Z_{t-1}$ and the deep transition feature $H_t^1$ through $S_t$, which allows the EPM to adaptively learn both short-term and long-term frame relations. In the causal LSTM of the EPM, the spatial memory is a function of the temporal memory through another set of gate structures. As the recurrence depth along the spatio-temporal transition paths grows considerably, each pixel in the final generated frame has a bigger receptive field of the input sequence at each time step, which is why the EPM has a better ability to model short-term video dynamics and sudden changes. Figure 5b displays the data flow process in the spatial memory of the EPM. The dimension of the input sequential satellite data (4 h and a time interval of 10 min) is $24 \times 32 \times 32$. At the first convolutional layer, the input terms $M_{n-1}^5$ and $C_n^1$ are concatenated to form a larger tensor, and then $H_n^1$ is generated by the convolution calculation. The EPM performs a total of five convolution calculations, with dimensions of hidden state from $10 \times 32 \times 32$ to $1 \times 32 \times 32$.

The EPM training process involves the use of the Adam optimizer with a learning rate of 0.001 and the mean square error as the loss metric. The input is a four-dimensional tensor of size [$c, t, h, w$] (6, 24, 32, 32), where $c$ represents the number of input channels, $t$ represents time (spanning 4 h), $h$ represents height, and $w$ represents width. Our research area focuses on PV plants, which provides us with images measuring $32 \times 32$ pixels (-128 km × 128 km). To train our model, we created a dataset consisting of sequences of 48 images for each channel, spanning 4 h. During the initial training process of the optimal network structure, data were obtained from resized AHI images at six bands over twelve manual observation stations and three all-sky imager stations. The training set utilized data from January to October in 2018, while the remaining months of 2018 were used as the validation set. In the application scenario of PV plants, the training method is described in the section of NRT and cyclically updated prediction system.

**Validation and assessment**. The primary metrics used to evaluate the accuracy of the NCP_CF system for forecasting the CF are the RMSE, MBE and R, defined as shown in Eqs. (19)-(21).

$$RMSE = \sqrt{\frac{1}{N} \times \sum_{i=1}^{N} (P_i - T_i)^2}, \quad (19)$$

$$MBE = \frac{1}{N} \times \sum_{i=1}^{N} (P_i - T_i), \quad (20)$$

$$R = \frac{\sum_{i=1}^{N}(P - \bar{P})(T_i - \bar{T})}{\sqrt{\sum_{i=1}^{N}(P_i - \bar{P})^2}\sqrt{\sum_{i=1}^{N}(T_i - \bar{T})^2}}, \quad (21)$$

where $P_i$ denotes the predicted CF, $T_i$ denotes the actual CF obtained from ground-based observations mentioned above, and $N$ is the total number of the matched samples.

## Reporting summary
Further information on research design is available in the Nature Portfolio Reporting Summary linked to this article.

## Data availability
The Himawari-8/9 data are available for free download from website [http://www.jma-net.go.jp/msc/en/]. The photos above PV plants are available for free exported from Google Earth Pro. Source data are provided with this paper.

## Code availability
Data processing, drawing and FCMA were conducted using PYTHON. Those codes can be accessed at [https://zenodo.org/doi/10.5281/zenodo.10148796]. The EPM code generated during the current study is available from the corresponding author upon request.

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

## Acknowledgements

We appreciate the Facebook Inc. for freely providing the Pytorch toolkit (https://pytorch.org). This study was supported partly by the Innovation Group Project of Southern Marine Science and Engineering Guangdong Laboratory (Zhuhai) (No.SML2023SP208), the National Natural Science Foundation of China (Grants 42175086 and U2142201), the FengYun Meteorological Satellite Innovation Foundation under Grant FY-APP-ZX-2022.0207, and the Science and Technology Planning Project of Guangdong Province (2023B1212060019).

## Author contributions

P.X. and M.M. conceived and designed the study. P.X., M.M., L.Z. and J.L. collaborated in discussing the results and writing the paper. Y.W. provided the actual power generation and surface solar radiation data. Y.Y. and S.J. provided the ground-based observation data. All authors contributed edits.

## Competing interests

The authors declare no competing interests.
