## [Peer review file · Nature Communications]

REVIEWER COMMENTS

Reviewer #1 (Remarks to the Author):

The paper in general has many good features and the method looks promising. The state of the article, as it stands now, needs revision (potentially major.)

Minor issues include geometry concerns for both the ground-based observation data (beginning lines 247) and for the Himawari cloud fraction data.

Regarding the ground observations, authors detail the EKO ASI-16 cloud fraction product, but do not specify how the cloud fraction at long limb angles from the 180° sensor are area-corrected. The optical geometry from the ASI-16 has significant compression at low angles and significant magnification at zenith angles - the accuracy of the derived cloud fraction will depend in great deal how this optical compression is corrected. More details on this process are needed.

Regarding the satellite observations, there are no details provided on parallax correction or terrain correction. Orthorectification of surface and cloud location is critical for computing accurate cloud fractions - failing to correct, or incorrectly performing the parallax or terrain correction will result in incorrectly-located clouds and inaccurate cloud fractions for a specific location (such as the three locations detailed in the paper.) More details on the methods of parallax and terrain correction, if performed, are needed (or a justification as to why they were not.)

A larger issue is the general layout of the paper - upon first read, the manuscript is confusing to put together. I would suggest a reorganization of the paper layout, thusly:

1.) Move section 4.) Methods (beginning line 234) to immediately after the introduction (starting line 95.) Move section 2.) Results (currently line 95) to after the end of the new Methods section, and then move 3.) Discussion to after the end of the new Results section. Going from Introduction -> Methods -> Results -> Discussion is a more natural way to read the paper - currently, the paper reads as Introduction -> Results -> Discussion -> Methods and is very confusing. This was perhaps a simple clerical error that could be addressed with minor updating of text and figure references.

The section detailing the PredRNN++ model is excellent, and if located earlier in the paper as suggested, helps bring the research together, and as stated before, the research presented is compelling and interesting. Minor details on ground and satellite geometry, as well as a reordering of the paper to flow better, are the recommendations here.

Reviewer #2 (Remarks to the Author):

1. A numeric tabular account of the RMSEs and R for the all the studied sites is needed to be able to observe and compare the effectiveness of the algorithm.

2. Line 155 – 157: “In order to better track the fast and stochastic changes in 156 cloud images, the neural network of the EPM is always cyclically generated by the 157 scheduled training process of the model, which has a one-hour update frequency.” Does the “cyclically generated” mean “finetuned based on the new set of images”? It is confusing. Would be it fair to assume that authors’ meant that the existing neural network is not deleted/removed instead the weight are updated based on the new data (images)? Is the finetuning being done online or offline? The “cyclically generated” point should be further explained.

3. Figure 4 doesn’t show the term ‘fast cloud mask’. It should be added to show where exactly is this masking algorithm being applied in the process.

4. It will be better to call out and address (as much as possible) the following caveats of using the proposed approach and algorithm:

- Vanishing gradient problem: when the forecast leading time exceeds 2 hours. The results show a reduction in the accuracy and stability of the predictions. This problem could be partly alleviated by the Gradient Highway architecture, which provides shorter routes for gradient flows, but it is still a challenge to capture complex and long-term variations in cloud images.
- Reliance on the quality of satellite data at visible and infrared bands: this data may be affected by low-visibility conditions, noise, or missing values. For example, according to Line 115 – 117, the invalid satellite visible images did contribute to moderate decrease in the forecast accuracy.
- Fast Cloud Mask Algorithm: The proposed system uses a fast cloud mask algorithm to determine the confidence value of each pixel, but this algorithm may or may not be able to handle all kinds of cloud types and situations. Either its robustness with respect to all cloud types, weather conditions (including wind, humidity and other atmospheric variables) need to test or this caveat needs to be called out.
- Generalizability claims: Need to bolster / support generalizability claims by including information about different regions as well as climates that have different cloud patterns and dynamics. Also, the article tests the model on two PV plants and several stations in China, but it does not compare its performance with other existing methods or datasets.

Reviewer #3 (Remarks to the Author):

This is a very interesting article on deterministic intraday solar satellite forecasting up to 4h ahead using advanced neural networks. Overall, I think the article has scientific merit, it should better frame its novelty in comparison to recent work in the field and can improve its evaluation framework. Also, there is some missing information and the article's structure is rather non-conventional. I provide the following feedback to the authors.

MAJORS

1. The algorithm's novelty or differences with similar work in the field is not clear. As far as I can tell, new features are the utilization of spectral satellite images (other than the visible channel) and the utilization of the past 4h images to forecast the next 4h (as usually only 2 to 4 previous images are used). However, from the algorithm itself, it is not clear to me if the proposal includes ad-hoc modifications to the neural network architecture to address the specific problem, or if it is a plain widely-used architecture adapted to the input-output problem (both cases are acceptable in my opinion, but should be clarified). In any case, differences, novelty, and/or similarities with respect to the current state-of-the-art architectures should be clearly given (Berthomier et al., 2020; Perez et al., 2021; Nielsen et al., 2021; Marchesoni-Acland et al., 2023; Garniwa et al., 2023).

<https://ieeexplore.ieee.org/document/9286606>

<https://www.sciencedirect.com/science/article/pii/S0038092X21001353>

<https://www.sciencedirect.com/science/article/pii/S0038092X21008306>

<https://www.sciencedirect.com/science/article/pii/S0038092X23004450>

<https://www.sciencedirect.com/science/article/pii/S0038092X23000439>

2. Training stage should be clarified. I understand the algorithm inputs the multi-spectral satellite radiances and outputs Cloud Fraction (CF). CF can be estimated by satellite or by ground cameras, and their comparison is non-trivial, say, there can be differences between both forms of estimation. So, is the algorithm trained to emulate the ground-measured CF or the satellite CF? Which is the ground truth for training? Then, the data set span is not given (6 months? 1 year?) and the train/test/validations splits are also not given. Or there is an online training? I recommend

commenting or highlighting all these issues.

3. Evaluation framework. Most of the evaluation is done by the correlation between the predicted CF with ground-measured CF, solar irradiance, and PV generation. Then, the RMSD for the CF comparison is also presented. I lacked the following:

- Scatter plots for predicted CF and ground-measured CF. Maybe a comparison between satellite and ground-measured CFs can provide some context as to which portion of the forecast uncertainty is fairly attributable to the forecast, and not the eventual mismatch between satellite and ground-measured CF. This is related to the previous point, which I maybe understood wrong. So, I recommend clarifying this issue, maybe authors can provide insights on these.
- In general, mean bias deviation, root mean squared error, and forecasting skills are required for comparison. Authors may check the recommended evaluation framework for the solar deterministic forecast(<https://www.sciencedirect.com/science/article/pii/S0038092X20303947>).
- Authors are very close to providing a solar radiation comparison. They just need a clear-sky model to convert the $CSR = (1-CF)$ to solar irradiance, by means of $SR = CSR * \text{Clear-Sky SR}$. This would allow them to compute metrics for the SR forecast, making it easier to compare with other works. Scatter plots are also recommended if authors decide to provide this comparison, which I think would add value to the article.
- I think the article is missing a comparison with the other works' metrics, to provide context to the performance findings.

MINORS

- It would be informative to include the word "satellite" in the title or keywords.
- Authors claim the performance downgrade after 2h ahead can be caused by the vanishing gradient problem. Just as a comment on this, in my opinion, this downgrade may be explained by the limited spatial domain that is considered (128 x 128 km). The size of the region limits the reachable forecast horizon. A cloud velocity of 30-50 km/h is roughly consistent with a 2h ahead forecast limitation at the center of a 128 km side square.
- Please indicate if a different neural network is trained for each forecast horizon or if there is another scheme for this.
- Line 258: It may be useful to provide an outline of the EKO cloud detection algorithm. This is related to point (2).
- Lines 297-299: There is a claim about a comparison between different architectures, but it does not say for which specific problem were the findings. Also, I would like to note that these metrics are not used in the solar forecasting field, as they typically are uninformative for this task (or, at least, there is no work demonstrating that are useful for the solar forecasting objectives).
- I wonder if the explanations of Lines 304-367 are useful to the article, as they maybe can be found in textbooks or original articles. On the other hand, I could not find the information regarding point (1) (differences with previous works or architectures).
- The article's structure is unusual, as metrics and models are given at last, and there is no conclusion section. Authors, of course, can choose how to present the article. I just point it out as a comment.

Reviewer #1 (Remarks to the Author):

The paper in general has many good features and the method looks promising. The state of the article, as it stands now, needs revision (potentially major.)

Response: Thank you very much for taking the time to review our manuscript. We greatly appreciate your constructive and kind comments. In response to your suggestions, we have provided a point-by-point response below:

Minor issues include geometry concerns for both the ground-based observation data (beginning lines 247) and for the Himawari cloud fraction data.

Regarding the ground observations, authors detail the EKO ASI-16 cloud fraction product, but do not specify how the cloud fraction at long limb angles from the 180° sensor are area-corrected. The optical geometry from the ASI-16 has significant compression at low angles and significant magnification at zenith angles - the accuracy of the derived cloud fraction will depend in great deal how this optical compression is corrected. More details on this process are needed.

Response: Thanks for your suggestion. Actually, we have added some sentences and further illustrated the cloud detection algorithm of EKO ASI-16 at Line 372 to 381 “*ASI, equipped with a digital camera coupled with an upward looking fisheye-lens, could provide field of view of about 180°, but pixels at a FOV >140 ° are excluded due to distortion. Digital images of the sky obtained by ASI are classified pixel by pixel into clear sky, optically thin and optically thick clouds, respectively. The cloud detection and opacity classification (CDOC) algorithm developed by Ghonima et al., 2012³² could provide 96%, 60%, and 96.3% accuracy in the validation for clear, thin, and thick cloud, respectively.*”. We believe the issue of optical compression you mentioned here is very important for large angle pixel. Please see our following answer to your question from the observation principles of the All-Sky Imager (ASI-16) and cloud fraction evaluation processes.

Fig 1

Firstly, all-sky imagers usually have a fisheye lens, which maps a hemispherical view field conformal to an image plane and produces a circular image. In the Ghonima’s cloud detection and opacity classification (CDOC) algorithm (Ghonima et al., 2012), pixels at a Field of view (FOV) $> 140^\circ$ are excluded due to distortion. ($\varphi > 70^\circ$ are excluded in above Fig 1). Hence, there is no area-correction for the cloud fraction at long limb angles from the 180° sensor.

In the CDOC algorithm, pixels in the images collected by the total/all sky imager are classified into three classes (clear, thin or thick) based on the difference between a pixel’s actual red-blue ratio (RBR) and the corresponding expected RBR if the pixel is clear. A haze correction factor (HCF) is added to account for the effects of variations in aerosol optical depth (AOD) on RBR. Compared to fixed thresholding technique, in the validation for clear, thin, and thick cloud, respectively, the CDOC algorithm (Ghonima et al., 2012) provided 96.0 %, 60.0 %, and 96.3 % accuracy as compared with 89.3 %, 56.1 %, and 91.5 %.

Secondly, **about image compression**, the explanation is as follows: In the case of an ideal fisheye lens, the image radius is directly proportional to the view angle, and the entire FOV is mapped onto concentric circles. However, in reality, a fisheye lens can introduce distortions, particularly for larger angles. This results in deviations from the linear relationship between the view angle and image radius, known as aberrations. Nonetheless, for the purpose of this discussion, we assume a linear and undistorted mapping. When using a fisheye lens for cloud monitoring, a cloud cover, which appears as a two-dimensional plane at a specific height in the sky, will be mapped onto a

hemispherical view field in the captured image, rather than maintaining its original planar shape (Sabburg et al., 2004; Kreuter et al., 2009).

Fig2

Fig3

Here is the imaging principle of a fisheye lens.

Fig2 shows that for mapping of a plane in constant height h to an angular dependent fisheye lens image, the distance d depends on the tangent of viewing angle. For zenith angles approaching 90° this distance will approach infinity. So the angular dependent concentric circles of the fisheye lens image for small angles φ maps the correspondent circle of the sky plane almost 1:1. For larger angles the mapped area will rise fast, until it even corresponds to infinity when approaching 90° .

Fig3 shows, how an existing fisheye lens image can be used to create a reproduction of the original sky plane by rear projection. Now the path of rays starts from the fisheye lens image and crosses an image plane in height a . The image plane covers the viewing angle ω . Dependent on ω and a , an image radius of R_ω results. The crossing point of image plane and angle φ results in height a and distance b and fulfils equations $h / d = a / b = \tan(\alpha)$. It is obvious to see, that path of rays produces a projected image ("Projection") that shows original sky plane true to scale.

In practice a constant viewing angle will be assigned to a constant image radius R_ω , whereby a constant distance a will result for all angles φ :

$$a / b = \tan(\alpha); \alpha = 90^\circ - \varphi \rightarrow a = b * \tan(90^\circ - \varphi)$$

By use of the specified values for ω and R_ω the constant a can be calculated:

$$a = R_\omega * \tan(90^\circ - \omega)$$

Using this it can be calculated, in which distance b results out of a projection angle φ . So every radius of the fisheye lens image results in a radius of the plane image. R_ω

sets the unit of measurement and it is most simple to use the unit "pixel". This way a pixel of the fisheye lens image can be moved to correspondent position on sky plane:

$$\mathbf{b} = \mathbf{a} / \tan(90^\circ - \varphi) = \mathbf{a} / \tan(90^\circ - 90^\circ * \mathbf{r}/\mathbf{R})$$

with \mathbf{r} equal to the current radius and \mathbf{R} equal to whole radius of fisheye lens image.

By using $\tan(\alpha) = \cot(90^\circ - \alpha)$, $\cot(\alpha) = 1/\tan(\alpha)$ and keeping periodicity of tangent in mind it is possible to simply the equation:

$$\mathbf{f}(\mathbf{r}) = \mathbf{a} * \tan(90^\circ * \mathbf{r}/\mathbf{R})$$

("forward calculation": Fisheye lens pixel to image plane)

This "forward calculation" doesn't include, that size of pixel raises in dependency from the angle, so projecting pixel one to one causes rising gaps between the pixels. A simple solution to avoid this problem is to calculate the pixel of the image plane backward to fisheye lens image and taking over one pixel for several times, when necessary:

$$\mathbf{f}(\mathbf{r})-1 = \mathbf{R} / 90^\circ * \text{atan}(\mathbf{b} / \mathbf{a})$$

("backward calculation": plane image pixel with distance \mathbf{b} to fisheye lens pixel with radius \mathbf{r})

This way a complete image will be created, but because of aliasing for large zenith angles there will result an aberration from the original sky plane. The pixel is not only enlarged, but also a jitter appears, because the fisheye lens pixel won't match to an even number of plane image pixel. For plane pixel located on borderline of two fisheye lens pixel an interpolation should be used to reduce aberration.

Limits of useful graphical representation:

To appraise the effect of rising zenith angle and size of aberration it makes sense to look at the underlying tangent function. Especially the derivation of the tangent

shows directly the difference between original fisheye lens pixel and resulting plane image pixel.

$f(r) = a * \tan(90^\circ * r/R)$ results in an angle range from 0° to 90° , because the ratio r/R covers the range from 0 to 1.

This plot shows the dependency of the plane image radius (y-axis) on zenith angle in degrees (x-axis). For angles less than 50° it is plain to see, that the radius rises almost linear with the angle, so the middle part of the image will project the original sky plane almost unaltered. In range from 50° to about 80° the projection will get more and more stretched until for angles larger than 80° the radius steeply rises to infinity. Also the derivation shows accordant factor of projection for range from 0° to 50° by 1:1 or 1:2 and fast rising for larger angles.

Reference:

J.M. Sabburg, C.N. Long: Improved sky imaging for studies of enhanced UV irradiance.

Atmos. Chem. Phys. Discuss., 4, 6213-6238, 2004.

A. Kreuter, M. Zangerl, M. Schwarzmann, M. Blumthaler: All-sky imaging: a simple, versatile system for atmospheric research. Applied Optics, Vol. 48, No. 7, 1. March 2009.

M.S. Ghonima, B. Urquart, C.W. Chow, J.E. Shields, A. Cazorla, J. Kleissl: A method for cloud detection and opacity classification based on ground based sky imagery. Atmos. Meas. Tech., 5, 2881-2892, 2012.

Regarding the satellite observations, there are no details provided on parallax correction or terrain correction. Orthorectification of surface and cloud location is critical for computing accurate cloud fractions - failing to correct, or incorrectly performing the parallax or terrain correction will result in incorrectly-located clouds and inaccurate cloud fractions for a specific location (such as the three locations detailed in the paper.) More details on the methods of parallax and terrain correction, if performed, are needed (or a justification as to why they were not.)

Response: Thank you for your suggestion. Indeed, geo-satellite products may encounter parallax issues when contrasted with ground-based information, potentially exerting a notable impact on the computation of precipitation stemming from low cloud-top temperature cloud clusters. Consequently, when investigating intense precipitation events or typhoons, it becomes imperative to take into account the parallax concerns associated with satellite products.

Nonetheless, prior research on the correction of cloud positions in satellite cloud images (Li et al., 2012) has underscored that the precision of parallax correction is intricately linked to the **accuracy of cloud-top height**. This challenge constitutes one of the principal explanations for the current scarcity of parallax correction in satellite full-disk image cloud parameter products. Regrettably, as of now, there is no precise and real-time Himawari-8 cloud-top height product with a spatial resolution of 4 km at our disposal for this investigation.

Another important reason for refraining from applying parallax correction to full-disk satellite products is that when a cloud envelops a considerably vast area and maintains relatively consistent features within a range akin to parallax correction (e.g., stratiform clouds), the influence of parallax correction tends to be relatively negligible (Wang et al., 2011). Certainly, this becomes particularly crucial when dealing with small clouds or when cloud characteristics exhibit rapid variation over short distances, or when both conditions coexist. Consequently, as previously stated, parallax correction is generally utilized for research endeavors pertaining to convection and typhoons, rather than for rectifying parallax in full-disk cloud products. **Hence, we firmly believe that any research focused on deep convection or typhoons necessitates the utilization of our product, and we think that conducting localized parallax correction is necessary.**

Furthermore, some previous studies have conducted numerous studies employing Himawari-8 data for cloud parameter inversion tasks, encompassing the retrieval of cloud optical thickness (COT), effective particle radius (CER), surface radiation (Letu et al., 2019; 2020). Nevertheless, it's worth noting that only a limited number of studies have incorporated parallax correction into satellite data preprocessing procedures. This might be due to the absence of high-precision cloud top height products. **Cloud top height products still manifest a notable degree of error, particularly in the case of relatively high cloud top samples with relatively larger retrieval errors (exceeding**

4 km when cloud top height exceeds 10 km, as illustrated in Figure 4 sourced from Min et al., 2020, see below Figure 1). Besides, the matched and nadir rectangle area for calculating cloud fraction is a 5×5 pixel box (more than 20×20 km), which is larger than the extreme parallax error of about 7~8 km (cloud top height must be >15 km, such as typhoon samples) (Li et al., 2012). Therefore, hastily implementing parallax corrections is likely to introduce some additional uncontrollable errors.

Fortunately, we find and refer to a most-related Chinese paper for the parallax correction of the future geostationary microwave sounder (Wei and Sun 2022, we have attached this paper in **Data Availability**). We think the theoretical studies in this paper can help us to further explain this issue. Figure 2 (see below) shows the sensitivity of position deviation (or parallax correction) to satellite imaging pixel spatial resolution, cloud top height (CTH), and view zenith angle using parallax correction algorithm (Wei and Sun 2022). First, it clearly proves that the position deviation is not sensitive to satellite imaging pixel spatial resolution. Second, the specific view zenith angles of ground-based stations used in this study from **Supplementary Table 2** are not larger than **60 degrees (the maximum value is about 56 degrees)**. This figure also indicates that the bias caused by parallax is unlikely to exceed 1km under the condition of view zenith angle < 60 degrees. Parallax correction should only be applied to exceptionally high clouds (>10km) resulting from typhoons or DCC with a view zenith angle greater than **70 degrees** in the GEO satellite field of view.

However, we have added two sentences to illustrate this issue at line 360 “*Besides, the view zenith angles of ground-based stations from H8/AHI field of view used here, as stated in Supplementary Table 2, are smaller than 60°. Therefore, the parallax effect is negligible (error < 1km) in the collocation between satellite pixels and ground-based stations for this study.*”

Figure 1. CTH retrieved from Himawari-8 satellite validation using CALIPSO product. (Figure 4 cited from Min et al., 2020)

Figure 2. Sensitivity of position deviation (or parallax correction) to satellite imaging pixel spatial resolution, cloud top height (CTH), and view zenith angle using parallax correction algorithm. (Figure 9 from Wei and Sun 2022).

References:

- Letu, H., Yang, K., Nakajima, T. Y., Ishimoto, H., Nagao, T. M., Riedi, J., Baran, A. J., Ma, R., Wang, T., Shang, H., Khatri, P., Chen, L., Shi, C., & Shi, J. (2020). High-resolution retrieval of cloud microphysical properties and surface solar radiation using Himawari-8/AHI next-generation geostationary satellite. *Remote Sensing of Environment*, 239, 111583. <https://doi.org/https://doi.org/10.1016/j.rse.2019.111583>
- Letu, Husi, T. M. Nagao, T. Y. Nakajima, J. Riedi, H. Ishimoto, A. J. Baran, H. Shang, M. Sekiguchi, and M. Kikuchi (2019): Ice cloud properties from Himawari-8/AHI next-generation geostationary satellite: Capability of the AHI to monitor the DC cloud generation process, *IEEE Transactions on Geoscience and Remote Sensing*, 57, 3229-3239, doi:10.1109/TGRS.2018.2882803.
- Li, W., Wang, H., Wu, Q., Yu, W., & Wang, Y. (2012). The Position Deviation of Geostationary Satellite Image and the Geometric Correction. *Acta Scientiarum Naturalium Universitatis Pekinensis*, 48(5).
- Wang, C., Luo, Z. J., & Huang, X. (2011). Parallax correction in collocating CloudSat and Moderate Resolution Imaging Spectroradiometer (MODIS) observations: Method and application to convection study. *Journal of Geophysical Research: Atmospheres*, 116(D17). <https://doi.org/https://doi.org/10.1029/2011JD016097>
- Min Min, Jun Li, Fu Wang, Zijing Liu, W. Paul Menzel, 2020. Retrieval of cloud top properties from advanced geostationary satellite imager measurements based on machine learning algorithms [J]. *Remote Sensing of Environment*, 239: 111616, doi: 10.1016/j.rse.2019.111616
- Xiaocheng Wei and Fenglin Sun, ANALYSIS OF THE PARALLAX CHARACTERISTICS OF GEOSTATIONARY- ORBITING MICROWAVE SOUNDER, *JOURNAL OF TROPICAL METEOROLOGY*, 2022, 38(6):901-914. (In Chinese)

A larger issue is the general layout of the paper - upon first read, the manuscript is confusing to put together. I would suggest a reorganization of the paper layout, thusly:

- 1.) Move section 4.) Methods (beginning line 234) to immediately after the introduction (starting line 95.)
- Move section 2.) Results (currently line 95) to

after the end of the new Methods section, and then move 3.) Discussion to after the end of the new Results section. Going from Introduction -> Methods -> Results -> Discussion is a more natural way to read the paper - currently, the paper reads as Introduction -> Results -> Discussion -> Methods and is very confusing. This was perhaps a simple clerical error that could be addressed with minor updating of text and figure references.

Response: Thanks for your suggestion which is very helpful for improving our manuscript. We sincerely apologize for any inconvenience caused to your review due to the format of our manuscript. Our format is as similar as possible to the official format of the journal that the manuscript will be published in. We will rearrange format according to the official requirements of the journal.

The section detailing the PredRNN++ model is excellent, and if located earlier in the paper as suggested, helps bring the research together, and as stated before, the research presented is compelling and interesting. Minor details on ground and satellite geometry, as well as a reordering of the paper to flow better, are the recommendations here.

Response: It is an excellent suggestion to include the details of the PredRNN++ model earlier in the paper. This will not only help to attract readers but also better showcase our research. Our format is as similar as possible to the official format of the journal that the manuscript will be published in. We will rearrange format according to the official requirements of the journal. Thank you once again for providing us with your valuable suggestions.

Besides, we also added more details on ground and satellite geometries at Line 402 “*about 20 km×20 km square area*” and at Line 364 “*each site would be centered around its precise location and matched with a 32×32 pixels box as the experimental area*”

Reviewer #2 (Remarks to the Author):

Response: Thank you very much for taking the time to review our manuscript. We greatly appreciate your constructive and kind comments. In response to your suggestions, we have provided a point-by-point response below:

1. A numeric tabular account of the RMSEs and R for the all the studied sites is needed to be able to observe and compare the effectiveness of the algorithm.

Response: Thanks for your suggestion. We have supplemented numeric tabular accounts of the metrics in our manuscript. It should be emphasized that we have changed some metrics used to evaluate the effectiveness of the algorithm in the new version of the manuscript. You can see the updated **Figure 2** and **Table S1** in the supplementary documentation.

Figure 2. RMSE and MBE between the predicted CF and the CF obtained by all-sky imagers. Subfigures (a), (c), (e) and (b), (d), (f) are the RMSEs and MBEs for Beijing, Nanjing and Zhuhai all-sky imager stations, respectively. The different colored lines represent the results for different months, and the dashed black line represents the mean of all the lines. Subfigure (g) depicts the mean RMSE and MBE of three all-sky imager sites at different local time of one day.

Supplementary Table 1. MBE and RMSE between cloud fraction prediction results and manual observations at different forecast horizon.

Station	1h		2h		3h		4h	
	MBE	RMSE	MBE	RMSE	MBE	RMSE	MBE	RMSE
Hailisu	0.03256	0.22325	0.02895	0.26958	-0.0012	0.31066	0.03862	0.34259
Liupanshan	-0.04537	0.24666	-0.05423	0.28695	-0.05407	0.32912	-0.04431	0.35657
Xilinhote	0.08632	0.26918	0.04201	0.30063	0.0752	0.34329	0.07927	0.37779
Changchun	0.02969	0.2767	0.00056	0.31907	0.00854	0.35496	0.00285	0.38825
Miyun	-0.01819	0.23676	-0.00595	0.29076	-0.00809	0.32905	0.02504	0.36319
Chengtoushan	-0.02679	0.23282	-0.03936	0.28175	-0.04419	0.33722	-0.01274	0.36239
Songshan	-0.01179	0.20495	-0.04906	0.25389	-0.05057	0.29714	-0.05801	0.34374
Xiaogan	0.09024	0.22986	0.07312	0.27931	0.06077	0.305	0.08475	0.32827
Youyang	-0.01127	0.16686	-0.04715	0.21286	-0.06077	0.24932	-0.06985	0.29385
Hangzhou	0.08225	0.22801	0.07797	0.28728	0.07855	0.31434	0.0737	0.3391
Chongwu	0.04786	0.24414	0.06828	0.30888	0.10071	0.35635	0.10416	0.38932
Zengcheng	0.06663	0.24232	0.03615	0.26559	0.04877	0.29168	0.06033	0.3244

2. Line 155 – 157: “In order to better track the fast and stochastic changes in 156 cloud images, the neural network of the EPM is always cyclically generated by the 157 scheduled training process of the model, which has a one-hour update frequency.” Does the “cyclically generated” mean “fine tuned based on the new set of images”? It is confusing. Would be it fair to assume that authors’ meant that the existing neural network is not deleted/removed instead the weight are updated based on the new data (images)? Is the fine tuning being done online or offline? The “cyclically generated” point should be further explained.

Response: Thanks for your suggestion. Actually, we have illustrated this issue at Line 161 “For instance, a cyclic training process of the model starts at the scheduled local time of 12:08 and ends at about 12:53, the imported training samples are from 08:00 to 12:00, and the latest updated EPM is timely activated for CF nowcasting at about

13:08 (nowcasting from 13:00 to 17:00) and 13:38 (nowcasting from 13:30 to 17:30), respectively, with an update frequency of half an hour. During the same period, the network of the next EPM is also trained simultaneously by using the sequential satellite images from 09:00 to 13:00. ” Indeed, the “cyclically generated” model is a fine tuned nowcasting model for **one** solar photovoltaic plant based on two GPU cards. It will update and replace the existing nowcasting model every hour, ensuring that we always have the most up-to-date nowcasting model. So the latest training dataset or satellite observation data will be used to train the new model, ensuring that it incorporates the most recent data available. This helps improve the accuracy of the predictions made by the model. This is a large and significant innovation of our work, combining practical application at PV plants and machine learning features. Note that, to make this paragraph more clear. We have added a sentence at the end of this paragraph to further explain this issue. *“This cyclical procedure will continually update and replace the existing EPM every hour, ensuring that we always have the most up-to-date nowcasting model.”*

As satellite data is continually updated, it is indeed important to update the training data of the model and also update the model itself. To address the need for rapid updates of real-time satellite observation data in CF nowcasting, we have developed a groundbreaking cyclically updated training method. This approach allows for timely incorporation of new data into the training process, ensuring that the model stays up-to-date with the most recent information available.

In the real application scenarios described in this manuscript (as we introduced here, we have used this novel method in 5 real solar PV plants belong to the CGN Wind Energy Co., LTD.), it is crucial to continuously update the local CF nowcasting model at every solar photovoltaic plant in order to maintain its prediction effectiveness. Additionally, we followed traditional methods and used a large amount of data for training, validation, and testing purposes when adjusting the parameters of the neural networks. These parameters here mainly refer to the Hyperparameter, such as learning rate, number of epochs et al.

The changes in clouds are very rapid and easily influenced by various meteorological factors, and they also have obvious regional and climatic characteristics. The prediction results of a unified model trained with a large amount of data and time are often affected by various sudden weather or climate conditions, resulting in unsatisfactory prediction results. Our method is to train and validate a set of optimal

hyperparameters through a certain amount of data in the early stage, and then train the model with the latest 4h satellite data in practical application scenarios. This model will be used for prediction in the next hour and will be used every 10 minutes (based on the update frequency of satellite data). As the data continues to update, the new model will be continuously trained, and each site's model will be trained based on its own data. This method can capture various mutation features in the data, ensuring maximum prediction performance for the next 4 hours.

3. Figure 4 doesn't show the term 'fast cloud mask'. It should be added to show where exactly is this masking algorithm being applied in the process.

Response: Thanks for your suggestion. 'Fast cloud mask algorithm (FCMA)' should indeed be indicated in Figure 4, and we have made changes to Figure 4 (see below figure) as required.

4. It will be better to call out and address (as much as possible) the following caveats of using the proposed approach and algorithm:

- Vanishing gradient problem: when the forecast leading time exceeds 2 hours. The results show a reduction in the accuracy and stability of the predictions. This problem could be partly alleviated by the Gradient Highway architecture, which provides shorter routes for gradient flows, but it is still a challenge to capture complex and long-term variations in cloud images.

Response: Thanks for your suggestion. Agree with your suggestion. The predictive ability of the model is indeed limited, and its predictive stability can fluctuate in different application scenarios; We realize that the predictive performance of the model will decrease with the increase of prediction time, so we have developed a cyclic prediction system and continuously updated it (see Q#2). This can indeed provide better predictive factors for photovoltaic power generation prediction in practical application scenarios of photovoltaic plants (we have tested it at five different PV plants in this study).

In addition, we predict that the time resolution of the product is 10 minutes (same as H8/AHI observation frequency), which not only includes discrete point data but also enables the display of more detailed information. However, it should be noted that this approach sacrifices some accuracy in exchange for a longer time prediction span.

When sudden changes happen (such as local convective cloud generation), future images should be generated upon nearby frames rather than distant frames, which requires that the predictive model learns short-term dynamics. If the model can better solve the problem of short-term mutation and long-term gradient disappearance, then the predictive ability will definitely further improve.

- Reliance on the quality of satellite data at visible and infrared bands: this data may be affected by low-visibility conditions, noise, or missing values. For example, according to Line 115 – 117, the invalid satellite visible images did contribute to moderate decrease in the forecast accuracy.

Response: Thanks for your suggestion. Regarding the quality of satellite data (H8/AHI), it is presented on the JMA official website https://www.data.jma.go.jp/mscweb/data/monitoring/gsics/ir/monit_geoleoir.html. It shows the very high quality of H8 GEO satellite multiband observation data. The H8 is the first operational satellite of the new-generation of GEO series in the world, which was launched in 2014, now it is replaced by H9 for operational application, H9 has the same imager and same orbital location of H8.

The quality of H8 GEO satellite data can be guaranteed in low-visibility conditions. Regarding missing values, we can provide full-disk data every 10 minutes for all 16 bands of H8/AHI. However, during the morning when the sun's altitude angle is low,

there may be some incomplete visible data. This is depicted in the figures below:

In this case, the predictive performance of the model will be affected by the visible band data before 07:00. But fortunately, the power generation of the solar photovoltaic station is basically close to zero in the morning, and the demand for the predictive ability of the model is not high during this period.

- **Fast Cloud Mask Algorithm:** The proposed system uses a fast cloud mask algorithm to determine the confidence value of each pixel, but this algorithm may or may not be able to handle all kinds of cloud types and situations. Either its robustness with respect to all cloud types, weather conditions (including wind, humidity and other atmospheric variables) need to test or this caveat needs to be called out.

Response: Thanks for your suggestion. Firstly, we would like to explain here why we need to develop the fast cloud mask algorithm. Operational cloud mask algorithm (OCMA) does consider various cloud types, surface conditions, and weather conditions, and is combined with a fast radiative transfer model and many other real-time ancillary data, such as numerical weather prediction data (see Wang et al., 2019). However, the cloud mask product of OCMA covers the entire satellite observation area (full-disk) and the specific process requires a relatively longer time and more financial resources. For 0-4 hour photovoltaic forecasting, timeliness is crucial. The cloud mask products only need to cover a small area directly above the photovoltaic plant, rather than the entire satellite observation area. Therefore, we have developed a new algorithm that can quickly and efficiently export local cloud mask products. Secondly, the FCMA

also considers various types of clouds and underlying surfaces, and it has been tested under various weather conditions. The following figure shows the climate and surface types of various sites during the FCMA testing process (Xia et al., 2023).

Station	Coordinate	Surface Type	Climate Type	Altitude
Gaolan	(103.95°E, 36.35°N)	Valley and basin	Temperate continental	1520 m
Beijing	(116.47°E, 39.80°N)	Plain	Warm temperate semi-humid and semi-arid monsoon	43.5 m
Changchun	(125.22°E, 43.90°N)	Plain	Temperate monsoon	300 m
Wuhan	(114.05°E, 30.60°N)	Hills	Subtropical monsoon	23.3 m
Hangzhou	(120.17°E, 30.23°N)	Hills	Subtropical monsoon	19 m
Shapingbai	(106.47°E, 29.58°N)	Hills and bench terrace	Subtropical monsoon humid	400 m
Guangzhou	(113.48°E, 23.22°N)	Middle and low mountains	Subtropical monsoon	4.2 m

Considering the cost and efficiency, we have developed the FCMA for scattered PV plants, which only works during daytime using six bands of H8/AHI (at 0.64, 0.86, 3.9, 7.0, 11.2, and 12.3 μm) and five inherited and improved cloudy/clear pixel tests from the MODIS official cloud mask algorithm (Frey et al., 2008). These five tests consider various types of clouds, which are very effective (that is also proved by our validation results compared to the ground-based CF observations).

In addition, several different sites utilize varying thresholds have already been test before. The threshold is adjusted to achieve optimal detection effectiveness. More details about FCMA can be found. in “Xia P, Min M, Yu Y, Wang Y, Zhang L. *Developing a near real-time cloud cover retrieval algorithm using geostationary satellite observations for photovoltaic plants. Remote Sensing* **15**, 1141 (2023).” Besides, we also present all the cloud detection thresholds corresponding to all sites in this manuscript in the **Code Availability section**. In Figure 1, we show aerial photos of each photovoltaic plant, which allows readers to see the surface type of each photovoltaic station more clearly.

Reference:

- Xi Wang, Min Min, Fu Wang, Jianping Guo, Bo Li, Shihao Tang, 2019. Intercomparisons of cloud mask product among Fengyun-4A, Himawari-8 and MODIS [J]. *IEEE Transactions on Geoscience and Remote Sensing*, 57(11): 8827-8839, doi: 10.1109/TGRS.2019.2923247
- Pan Xia, Min Min*, Yu Yu, Yun Wang, Lu Zhang, 2023. Developing a near real-time cloud cover retrieval algorithm using geostationary satellite observations for photovoltaic plants [J]. *Remote Sensing*, 15(4), 1141, doi: 10.3390/rs15041141

Frey, R.A.; Ackerman, S.A.; Liu, Y.; Strabala, K.I.; Zhang, H.; Key, J.R.; Wang, X. Cloud detection with MODIS. Part I: Improvements in the MODIS cloud mask for collection 5. *J. Atmos. Ocean. Technol.* 2008, 25, 1057–1072. [CrossRef]

- Generalizability claims: Need to bolster / support generalizability claims by including information about different regions as well as climates that have different cloud patterns and dynamics. Also, the article tests the model on two PV plants and several stations in China, but it does not compare its performance with other existing methods or datasets.

Response: Thanks for your valuable suggestion. Agree with your suggestion. Generalizability claims is indeed of great significance for the practicality of our machine-learning-based algorithm. The manual observation stations, all-sky imagers stations, and photovoltaic plants in our research have a large latitude and longitude span, which could demonstrate the generalization of our algorithm. Sites distributed at different latitudes could represent different climate types, cloud features.

For different PV plant sites, the EPM for CF nowcasting model will continually update and replace the existing EPM every hour (see Q#2), which can prove the generalization of our method. To further demonstrate the generalizability of our developed method in this study, we have incorporated **three additional PV plants** affiliated with China General Nuclear Power Group (CGN) Wind Energy Co Ltd (It is very difficult to obtain business data from business company). These new PV plants include Xiaochengzi, Lijiamen, and Shiziyan. We conducted tests using data from each plant for a duration of one month (same as the two PV plants mentioned before in the manuscript). The updated testing outcomes are presented in the updated Figure 3, while the locations of the new plants can be observed in the updated Figure 1.

Figure 3. Power and GTI of PV plants and corresponding 0-4h CSR. Time series of the power (MW), GTI ($W \cdot m^{-2}$) and predicted clear sky ratio (CSR) at (a) Sangge, (b) Leling, (c) Xiaochengzi, (d) Lijiamen, and (e) Shiziyuan PV plants from 09:00 to 17:00 (local time or Beijing time) on each day in November of 2022. R1–R4 and R5–R8 indicate the Rs (correlation coefficient) of the predicted CSR with the power and the GTI for forecast leading time of 1–4 hours, respectively. Note that the missing and invalid data are not shown. The results for other months are shown in Figs. S3–S6 of the supplementary materials.

Figure 1. Geographical distributions and photographs of ground-based sites. PV plants (dark green small solid circles), twelve manual observation stations (red solid circles), three all-sky imager stations (blue solid circles) and five PV power test plants (yellow solid circles).

Reviewer #3 (Remarks to the Author):

Response: Thank you very much for taking the time to review our manuscript. We greatly appreciate your constructive and kind comments. In response to your suggestions, we have provided a point-by-point response below:

This is a very interesting article on deterministic intraday solar satellite forecasting up to 4h ahead using advanced neural networks. Overall, I think the article has scientific merit, it should better frame its novelty in comparison to recent work in the field and can improve its evaluation framework. Also, there is some missing information and the article's structure is rather non-conventional. I provide the following feedback to the authors.

MAJORS

1. The algorithm's novelty or differences with similar work in the field is not clear. As far as I can tell, new features are the utilization of spectral satellite images (other than the visible channel) and the utilization of the past 4h images to forecast the next 4h (as usually only 2 to 4 previous images are used). However, from the algorithm itself, it is not clear to me if the proposal includes ad-hoc modifications to the neural network architecture to address the specific problem, or if it is a plain widely-used architecture adapted to the input-output problem (both cases are acceptable in my opinion, but should be clarified). In any case, differences, novelty, and/or similarities with respect to the current state-of-the-art architectures should be clearly given (Berthomier et al., 2020; Perez et al., 2021; Nielsen et al., 2021; Marchesoni-Acland et al., 2023; Garniwa et al., 2023).

<https://ieeexplore.ieee.org/document/9286606>

<https://www.sciencedirect.com/science/article/pii/S0038092X21001353>

<https://www.sciencedirect.com/science/article/pii/S0038092X21008306>

<https://www.sciencedirect.com/science/article/pii/S0038092X23004450>

<https://www.sciencedirect.com/science/article/pii/S0038092X23000439>

Response: Thanks for your suggestion and references. We have learned a lot from these papers. In this study, the novelty of our algorithm lies in its direct utilization of Level

1B (L1B) radiance data from H8/AHI. We use both visible and infrared channels as input for the EPM model, which then predicts the Level-1B radiance data for the next 4 hours. Additionally, we have developed a fast cloud mask algorithm that, when combined with the predicted radiance data, allows us to obtain cloud masks and fractions specifically for solar photovoltaic plants.

Compared with the work of Franco Marchesoni-Acland et al., (2023), they only use a visible channel, and their cloud detection principle is that visible channel typically reflects more sunlight than the ground, appearing brighter in the images. This assumption does not hold for high albedo terrains, containing snow or salt flats. Our method used six channels (two visible and four infrared) of AHI and developed a relatively comprehensive and adaptable cloud detection method that can detect different types of clouds under different and variable conditions (refer to U.S. NASA MODIS official cloud mask algorithm). It is practical and more complex in environmental conditions, and the detection effect is also more accurate. In practical application scenarios at PV plants, our new method provides timeliness and seamless integration, which is of great importance in the context of CF nowcasting at the next 0-4 hours (refer to Reviewer-2 Q#2 for a detailed flowchart depicting the specific process of this novel method).

Besides, Predrnn++ is a plain widely-used architecture adapted to the input-output problem. To achieve greater spatio-temporal modeling capability, we set the EPM with a six-layer architecture (five convolutional layers and one GHU layer). Based on the spatial size and temporal resolution of the data, we carefully design the dimensions of temporal memory (C_n^k) and spatial memory (M_n^k) in each convolutional layer. The advantages of GHU and causal LSTM have explained at lines 489-498. “*The gradient highway (blue line in Fig. 5a) supplies a quick path from the first to the last time step by quickly updating hidden state Z_t . It is worth noting that, unlike temporal skip connections, the GHU controls the proportions of Z_{t-1} and the deep transition feature H_t^1 through S_t , which allows the EPM to adaptively learn both short-term and long-term frame relations. In the causal LSTM of the EPM, the spatial memory is a function of the temporal memory through another set of gate structures. As the recurrence depth along the spatio-temporal transition paths grows considerably, each pixel in the final generated frame has a bigger receptive field of the input sequence at each time step,*

which is why the EPM has a better ability to model short-term video dynamics and sudden changes. ”

Due to the distinct characteristics of data obtained from various ground-based sites, our model prioritizes both speed and accuracy. Consequently, we have refrained from training a single unified model for different sites. Instead, each PV (photovoltaic) plant undergoes individualized training using its own specific data, thereby creating a highly tailored model that exhibits superior predictive performance at its respective location. This approach ensures optimal targeting and enhances the overall prediction capabilities of the model. This cyclical updates in our prediction model represent a substantial and noteworthy innovation within our work. This approach combines the practical application of PV plants with advanced machine learning capabilities, resulting in a powerful synergy.

Reference:

Franco Marchesoni-Acland, Andrés Herrera, Franco Mozo, Ignacio Camiruaga , Alberto Castro , Rodrigo Alonso-Suárez; Deep learning methods for intra-day cloudiness prediction using geostationary satellite images in a solar forecasting framework, *Solar Energy* 262 (2023) 111820.

2. Training stage should be clarified. I understand the algorithm inputs the multi-spectral satellite radiances and outputs Cloud Fraction (CF). CF can be estimated by satellite or by ground cameras, and their comparison is non-trivial, say, there can be differences between both forms of estimation. So, is the algorithm trained to emulate the ground-measured CF or the satellite CF? Which is the ground truth for training? Then, the data set span is not given (6 months? 1 year?) and the train/test/validations splits are also not given. Or there is an online training? I recommend commenting or highlighting all these issues.

Response: Thanks for your suggestion. We have added a paragraph to clarify training stage. *“The EPM training process involves the use of the Adam optimizer with a learning rate of 0.001 and the mean square error as the loss metric. The input is a four-dimensional tensor of size $[c, t, h, w]$ (6, 24, 32, 32), where c represents the number of*

input channels, t represents time (spanning 4 hours), h represents height, and w represents width. Our research area focuses on PV plants, which provides us with images measuring 32×32 pixels (approximately $128\text{km} \times 128\text{km}$). To train our model, we created a dataset consisting of sequences of 48 images for each channel, spanning 4 hours. During the initial training process of the optimal network structure, data were obtained from resized AHI images at six bands over 12 manual observation stations and 3 all-sky imager stations. The training set utilized data from January to October in 2018, while the remaining months of 2018 were used as the validation set. In the application scenario of PV plants, the training method is described in the section of NRT and cyclically updated prediction system.” Other questions are answered below.

(1) Firstly, we have thoroughly illustrated and clarified the detailed training stage in our response to Reviewer-2 Q#2. We believe this is a crucial aspect of our study. Also, we have emphasized this issue at Line 234 to highlight its significance.

(2) Secondly, The cloud fraction (CF) obtained from ground-based stations is considered as the reference truth in our study. We have validated the retrieved results from GEO satellite imagers through manual observations and the ASI-16 method. Besides, the approach we utilized to derive cloud fraction (CF) from GEO satellites does not rely on CF measurements obtained through ground-based methods. Instead, we employ a training process using continuous radiance images captured by GEO satellites to extract cloud masks and calculate cloud fraction.

(3) Thirdly, our algorithm is trained to emulate the satellite data, and the ground truth for training is satellite data. The nowcasting CF is retrieved by FCMA based on the predicted satellite image data.

Due to the differences between our training method and traditional methods, our initial training was only to find the best network architecture parameters (the number of layers, epoch, and learning rate etc.) for the model. Satellite data from Jan. to Oct. in 2018 at 12 manual observation stations and 3 all-sky imager stations were used as training set and the others in 2018 were used as validation set. In this process, we mainly saved the optimal parameters of EPM in terms of structure. **In the subsequent application within the system depicted in Figure 4, we employed online training to continuously update the model, thereby ensuring the stability of its predictive capabilities** (also refer to Reviewer-2 Q#2 for a detailed flowchart depicting the specific process of this novel method).

Note that different sites will train different models based on their own data, and each model is only used for prediction in the next 1 hour. As satellite data updates, new models will be continuously trained. This method aims to capture short-term changes in cloud fraction from 0 to 4 hours at each site and provide better prediction results. The traditional method of using a large amount of data to train a unified and unchanging model is difficult to capture rapid cloud changes in small areas above photovoltaic stations, so this method is not adopted.

3. Evaluation framework. Most of the evaluation is done by the correlation between the predicted CF with ground-measured CF, solar irradiance, and PV generation. Then, the RMSD for the CF comparison is also presented. I lacked the following:

- Scatter plots for predicted CF and ground-measured CF. Maybe a comparison between satellite and ground-measured CFs can provide some context as to which portion of the forecast uncertainty is fairly attributable to the forecast, and not the eventual mismatch between satellite and ground-measured CF. This is related to the previous point, which I maybe understood wrong. So, I recommend clarifying this issue, maybe authors can provide insights on these.

Response: Thanks for your advice. We plan to create scatter plots comparing predicted CF values with ground-measured CF values. However, the manual observed CF values only have one decimal place (e.g., 0.1, 0.2, 0.3, etc.), which might result in strange-looking scatter plots (see below figure).

The unit of the color-bar is the total number of values taken.

For the comparison between predicted CF and All-sky imager observed CF, the scatter plot is as follow:

However, this is indeed a shortcoming of our paper, so we have updated the evaluation parameters, redesigned Figure 2, and added Supplementary Table 1, hoping that those changes could better show the difference between the predicted CF and ground-measured CF.

As you said, there are indeed two parts of the error here. One comes from the prediction error of EPM, and the other comes from the error between the CF retrieved by satellite data and observed on the ground. We have made a detailed comparison between manual observed CF and satellite retrieved CF in our previous work and proven the error between them is small. “*Xia P, Min M, Yu Y, Wang Y, Zhang L. Developing a near real-time cloud cover retrieval algorithm using geostationary satellite observations for photovoltaic plants. Remote Sensing 15, 1141 (2023)*”. Here, our primary focus lies in determining whether the prediction outcomes of our model can effectively aid in forecasting photovoltaic power for the future 0-4 hour timeframe at the PV plant. In this context, we place greater emphasis on evaluating the overall system error depicted in Figure 4, rather than dissecting it into the two aforementioned components. By consolidating the evaluation of errors, we gain a comprehensive understanding of the model's performance and its potential to provide valuable assistance in accurate photovoltaic power forecasting within the specified time range.

- In general, mean bias deviation, root mean squared error, and forecasting

skills are required for comparison. Authors may check the recommended evaluation framework for the solar deterministic forecast

(<https://www.sciencedirect.com/science/article/pii/S0038092X20303947>).

Response: Thanks for your suggestion and reference. We have realized that the correlation coefficient (R) was not a good choice when evaluating the model prediction, so we added MBE, removed R and redesigned Figure 2 and added Supplementary Table S1(see below). Forecasting skill are often used to compare the forecasting effect between different forecasting models. Unfortunately, there is no standard CF forecasting model for comparison.

Figure 2. RMSE and MBE between the predicted CF and the CF obtained by all-sky imagers. Subfigures (a), (c), (e) and (b), (d), (f) are the RMSEs and MBEs for Beijing, Nanjing and Zhuhai all-sky imager stations, respectively. The different colored lines represent the results for different months, and the dashed black line represents the mean of all the lines. Subfigure (g) depicts the mean RMSE and MBE of three all-sky imager sites at different local time of one day.

Supplementary Table 1. MBE and RMSE between cloud fraction prediction results and manual observations at different forecast horizon.

Station	1h		2h		3h		4h	
	MBE	RMSE	MBE	RMSE	MBE	RMSE	MBE	RMSE
Hailisu	0.03256	0.22325	0.02895	0.26958	-0.0012	0.31066	0.03862	0.34259
Liupanshan	-0.04537	0.24666	-0.05423	0.28695	-0.05407	0.32912	-0.04431	0.35657
Xilinhote	0.08632	0.26918	0.04201	0.30063	0.0752	0.34329	0.07927	0.37779
Changchun	0.02969	0.2767	0.00056	0.31907	0.00854	0.35496	0.00285	0.38825
Miyun	-0.01819	0.23676	-0.00595	0.29076	-0.00809	0.32905	0.02504	0.36319
Chengtoushan	-0.02679	0.23282	-0.03936	0.28175	-0.04419	0.33722	-0.01274	0.36239
Songshan	-0.01179	0.20495	-0.04906	0.25389	-0.05057	0.29714	-0.05801	0.34374
Xiaogan	0.09024	0.22986	0.07312	0.27931	0.06077	0.305	0.08475	0.32827
Youyang	-0.01127	0.16686	-0.04715	0.21286	-0.06077	0.24932	-0.06985	0.29385
Hangzhou	0.08225	0.22801	0.07797	0.28728	0.07855	0.31434	0.0737	0.3391
Chongwu	0.04786	0.24414	0.06828	0.30888	0.10071	0.35635	0.10416	0.38932
Zengcheng	0.06663	0.24232	0.03615	0.26559	0.04877	0.29168	0.06033	0.3244

- Authors are very close to providing a solar radiation comparison. They just need a clear-sky model to convert the CSR = (1-CF) to solar irradiance, by means of $SR = CSR * \text{Clear-Sky SR}$. This would allow them to compute metrics for the SR forecast, making it easier to compare with other works. Scatter plots are also recommended if authors decide to provide this comparison, which I think would add value to the article.

Response: Thanks for your suggestion. The solar radiation measured by photovoltaic plants is the radiation captured on solar photovoltaic panels, which is called global tilted irradiance (GTI). GTI can be estimated from global horizontal irradiance (GHI). With your inspiration, we are in the process of utilizing radiation measurements under cloudless conditions along with our CSR (Cloud Sky Rate) model to estimate the amount of radiation that reaches the surface at solar PV plants. We utilized the clear sky solar radiation product derived from the classical ERA5 reanalysis data (<https://cds.climate.copernicus.eu/cdsapp#!/dataset/reanalysis-era5-single-levels>) that has been spatially aligned with our specific solar PV plant location. **Mean surface**

downward short-wave radiation flux at clear sky (msdswrfcs) of ERA5 is the amount of solar radiation (also known as shortwave radiation) that reaches a horizontal plane at the surface of the Earth, assuming clear-sky (cloudless) conditions. We use $\text{msdswrfcs} * \text{CSR}$ as the GHI, and analyze its correlation with GTI and PV power generation. The comparison of Sangge station is shown as follow:

The time for radiation data is from 09:00 to 17:00 every day. The pattern of GTI is relatively stable, with the strongest radiation at noon and lower in the morning and evening. However, **msdswrfcs** fluctuates greatly throughout the day, and the patterns of different dates vary. This may be due to the data from ERA5 is an averaged value in a fixed region. And the spatial resolution of the ERA5 data is relatively low, making it difficult to directly obtain data near the specific photovoltaic plant, and data can only be obtained through deviation calculation. We use $\text{CSR} * \text{msdswrfcs}$ as the solar radiation reaching the surface of PV plant (GHI), and then compare its change trend with the measured GTI of the power plant, it is found that the change trend of GHI was influenced by **msdswrfcs** and fluctuated greatly in a day, which is quite different from the change trend of GTI.

The comparison of Lelin station is shown as follow:

Due to the large fluctuations in the distribution of msdwsrxfc data throughout the day, it is difficult for us to combine this radiation data to estimate the GHI on the surface of photovoltaic stations and compare it with GTI. **It seems better to directly use our predicted CSR to compare with GTI.**

- I think the article is missing a comparison with the other works' metrics, to provide context to the performance findings.

Response: Thanks for your suggestion. We know exactly what you mean, but it's a tough thing for us to achieve it. Our research is conducted in conjunction with practical application scenarios of PV plants, which makes our work very different from other works. For the CF prediction of the next 4 hours, the data from the first 4 hours is often very important. Our method is to directly use L1B data, which ensures the timeliness of the data. Our goal is to predict the precise changes in cloud fraction of the next 4 hours over PV plants. Unlike many methods that use visible images (Franco Marchesoni-Acland et. al, 2023; Pranda M.P. Garniwa et. al, 2023;) to obtain cloud fractions, we have developed a fast cloud detection algorithm that uses a total of 6

channels satellite data and considers different surface types. This method can quickly and accurately retrieve cloud fractions. In addition, many previous works have a relatively large experimental scope (1024km ×1024km in Franco Marchesoni-Acland et. al, 2023; squares of size 256 × 256 pixels showing France and the final resolution of the images was 4.5 kilometers in Lea Berthomier et. al, 2020;). In order to better capture cloud changes near PV plants and save computational resources, our experimental scope is relatively small (128km×128km), which makes it difficult to compare with other works.

Regarding the predictive ability of neural networks, we also hope to demonstrate the feasibility of our method through Forecast Skill, but no one has adopted our method in previous work. The differences in model data input and cloud detection methods make it difficult to compare. But when we first chose the neural network structure, we consulted a lot of information and chose a model (PredRNN++, Yunbo Wang et. al, 2018) with strong prediction ability that is suitable for our prediction task. Our EPM has been running at five independent PV plants (see Figure 1) for about half a year (started at Nov. 2022), and it has been proven that its predicted CF can greatly improve the accuracy of photovoltaic power generation prediction (see Figure 3).

Reference:

- Franco Marchesoni-Acland, Andrés Herrera, Franco Mozo, Ignacio Camiruaga , Alberto Castro , Rodrigo Alonso-Suárez; Deep learning methods for intra-day cloudiness prediction using geostationary satellite images in a solar forecasting framework, *Solar Energy* 262 (2023) 111820.
- Pranda M.P. Garniwa , Rial A. Rajagukguk , Raihan Kamil , HyunJin Lee; Intraday forecast of global horizontal irradiance using optical flow method and long short-term memory model, *Solar Energy* 252 (2023) 234–251.
- Lea Berthomier, Bruno Pradel,Lior Perez; Cloud Cover Nowcasting with Deep Learning, 978-1-7281-8750-1/20/\$31.00 ©2020 IEEE.

Wang Y, Gao Z, Long M, Wang J, Yu PS. PredRNN++: Towards a resolution of the deep-in-time dilemma in spatiotemporal predictive learning. In: Proceedings of the 35th International Conference on Machine Learning. PMLR (2018).

MINORS

- It would be informative to include the word "satellite" in the title or keywords.

Response: Per your suggestion, we have changed the title of this paper to “**Accurate nowcasting of cloud cover at solar photovoltaic plants using geostationary satellite images**”

- Authors claim the performance downgrade after 2h ahead can be caused by the vanishing gradient problem. Just as a comment on this, in my opinion, this downgrade may be explained by the limited spatial domain that is considered (128 x 128 km). The size of the region limits the reachable forecast horizon. A cloud velocity of 30-50 km/h is roughly consistent with a 2h ahead forecast limitation at the center of a 128 km side square.

Response: Thanks for your suggestion. In the specified range of 128km×128km (which is actually about 150km×150km or larger due to the nominal 4km spatial resolution being applicable only to the nadir point pixel). Cloud movement probably affects the forecasting performance of EPM after 2 hours. However, we believe that this impact is relatively small. (1) Firstly, our EPM takes into account the spatial changes in cloud image features when making predictions. This consideration reduces the influence of cloud movement on prediction accuracy. We also cyclically update our EPM every hour using multiple continuous satellite images. This frequent updating can help compensate for the effects of cloud movement and improve the overall forecasting performance. (2) Secondly, we have made multiple attempts to determine the final size of the prediction region, considering factors such as available computer resources, model computation time, and test results. After careful evaluation, we settled on a 32×32 pixel box. We experimented with larger sizes (such as 64×64 pixel box (cost 4 hours for one training

session) and 128×128 pixel box), but ultimately found that 128km×128km provided a good balance between computational efficiency and accurate predictions.

The rapid movement of clouds may indeed worsen the prediction results, but we believe that this impact is relatively small here. Thank you again for your valuable suggestions. In our revised manuscript, we add the potential reason of the decrease in prediction performance caused by cloud movement at line 275-276, “ *By the limited spatial domain, the rapid movement of clouds may cause a small bias between the predicted CF and the actual CF.* ”

- Please indicate if a different neural network is trained for each forecast horizon or if there is another scheme for this.

Response: Thanks for your suggestion. We employed a fixed neural network to train all prediction models for various forecast horizons. Additionally, the model used for making predictions is updated on an hourly basis. At Line 424, we have depicted the PredRNN++ model, which relies on a framework utilizing recurrent neural networks (RNNs) with long short-term memory (LSTM). This model has the capability to forecast future CF within the range of 0 to 4 hours.

- Line 258: It may be useful to provide an outline of the EKO cloud detection algorithm. This is related to point (2).

Response: Thanks for your suggestion. The EKO cloud detection algorithm is the cloud detection and opacity classification (CDOC) algorithm. More details could be found in “*Ghonima et al., 2012.*” In the CDOC algorithm, pixels in the images collected by the total/all sky imager are classified into three classes (clear, thin or thick) based on the difference between a pixel’s actual red-blue ratio (RBR) and the corresponding expected RBR if the pixel were clear. A haze correction factor (HCF) is added to account for the effects of variations in aerosol optical depth (AOD) on RBR. Compared to fixed thresholding technique, in the validation for clear, thin, and thick cloud, respectively, the CDOC algorithm provided 96.0 %, 60.0 %, and 96.3 % accuracy as compared with 89.3 %, 56.1 %, and 91.5 % (Ghonima et al., 2012).

Reference:

Ghonima MS, Urquart B, Chow CW, Shields JE, Cazorla A, Kleissl J. A method for cloud detection and opacity classification based on ground based sky imagery. *Atmospheric Measurement Techniques* 5, 2881–2892, 2012.

- Lines 297-299: There is a claim about a comparison between different architectures, but it does not say for which specific problem were the findings. Also, I would like to note that these metrics are not used in the solar forecasting field, as they typically are uninformative for this task (or, at least, there is no work demonstrating that are useful for the solar forecasting objectives).

Response: Thanks for your suggestion. The comparison here mainly analyzes the advantages of different structures in dealing with spatiotemporal sequence problems. The comparison here does not involve specific problems for analysis, but mainly demonstrates the structural advantages of PredRNN++ compared to other networks.

About metrics, we have realized that the correlation coefficient (R) was not a good choice when evaluating the model prediction, so we added MBE, removed R and redesigned Figure 2 and added Table S1 in the supplementary documentation.

- I wonder if the explanations of Lines 304-367 are useful to the article, as they maybe can be found in textbooks or original articles. On the other hand, I could not find the information regarding point (1) (differences with previous works or architectures).

Response: Thanks for your suggestion. We place this explanation here mainly to facilitate readers to have a clearer understanding of the PredRNN++ structure in Figure 5, and to be able to clearly see the data transfer in each memory. The main structural change is that we set the number of convolutional layers to 5, and the data dimensions of each layer's memory unit have been carefully designed, as shown in Figure 5B.

- The article's structure is unusual, as metrics and models are given at last, and there is no conclusion section. Authors, of course, can choose how to present the article. I just point it out as a comment.

Response: Thanks for your suggestion. We sincerely apologize for any inconvenience caused to your review due to the format of our manuscript. Our format is as similar as

possible to the official format of the journal that the manuscript will be published in.
We will rearrange format according to the official requirements of the journal.

REVIEWERS' COMMENTS

Reviewer #1 (Remarks to the Author):

Thank you for your edits and additions to the paper - it appears that many of the issues regarding parallax and pixel-matching between the ASI imager and satellite imagery have been addressed. Much of the supporting documentation that was provided in the 'response to reviewer' document could be included in the manuscript itself to support the assertions - this reviewer is satisfied that parallax concerns and imaging issues have been addressed, but readers without the benefit of the supporting material may not be.

I still find the layout of the paper confusing; with results first and discussion second, followed by methods, I still find myself flipping back and forth between the paper to interpret the results. This is a choice for the editor and the authors to consider, of course.

Reviewer #2 (Remarks to the Author):

Thanks for thoroughly addressing the comments and concerns from the previous round of review. No further suggestions from my end.

Reviewer #3 (Remarks to the Author):

I think the authors have made a good revision work. The reviewers' response has cleared all my concerns about the manuscript. I agree with the authors about not providing a tilted plane solar irradiance evaluation, which includes further considerations than the forecast itself. Most PV plants also record the global horizontal irradiance (GHI), and my previous comment referred to this magnitude. I also agree with not providing the discrete-values scatter plots, as they are uninformative. The non-inclusion of the forecasting skill metric, however, is debatable, as it can be easily constructed from CF persistence. Maybe I am losing something here, but the authors' reply on this issue was not clear to me.

My only comment left is that the reviewers' response frames the article better in the field than the current manuscript, in which there is no mention of previous satellite solar or cloud nowcasting methods based on neural networks. By this, it may be interpreted that this is the only solar satellite nowcasting work based on satellite images and neural networks, which it is not. In my opinion, the current manuscript's introduction reads as disconnected from the specific state-of-the-art.

Reviewer #1 (Remarks to the Author):

Thank you for your edits and additions to the paper - it appears that many of the issues regarding parallax and pixel-matching between the ASI imager and satellite imagery have been addressed. Much of the supporting documentation that was provided in the 'response to reviewer' document could be included in the manuscript itself to support the assertions - this reviewer is satisfied that parallax concerns and imaging issues have been addressed, but readers without the benefit of the supporting material may not be.

Response: Thanks for your valuable suggestion. We have added a sentence at line 333 “*For more explanations and details, please refer to Supplementary Note and Supplementary Figs. 7 and 8*”. In supplementary file, we have added Supplementary Note and Supplementary Figs. 7 and 8 to explain this parallax issue, which is written as follows:

***Supplementary Note:** We think the theoretical studies from Reference-1 (Wei and Sun 2022) mentioned above can help us to explain the relatively small or negligible parallax effect. Supplementary Figure 7 shows the sensitivity of position deviation (or parallax correction) to satellite imaging pixel spatial resolution, cloud top height (CTH), and view zenith angle using parallax correction algorithm (Wei and Sun 2022). First, it clearly proves that the position deviation is not sensitive to satellite imaging pixel spatial resolution. Second, the specific view zenith angles of ground-based stations used in this study from Supplementary Table 2 are not larger than 60 degrees (the maximum value is about 56 degrees). This figure also indicates that the bias caused by parallax is unlikely to exceed 1km under the condition of view zenith angle < 60 degrees. Parallax correction should only be applied to exceptionally high clouds (>10km) resulting from typhoons or Deep Convective Clouds with a view zenith angle greater than 70 degrees in the GEO satellite field of view.*

In addition, it is worth noting that the parallax correction for GEO satellite images needs high-precision cloud top height products. Cloud top height products still manifest a notable degree of error, particularly in the case of relatively high cloud top samples with relatively larger retrieval errors (exceeding 4 km when cloud top height exceeds

10 km, as illustrated in Supplementary Figure 8). This may introduce more uncontrollable errors to the parallax correction algorithm.

Supplementary Figure 7. Sensitivity of position deviation (or parallax correction) to satellite imaging pixel spatial resolution, cloud top height (CTH), and view zenith angle using parallax correction algorithm. (Supplementary Figure 7 from Reference-2: Xiaocheng Wei and Fenglin Sun, Analysis of the parallax characteristics of geostationray-orbiting microwave sounder [J]. Journal of Tropical Meteorology, 2022, 38(6): 901-914, doi: 10.16032/j.issn.1004-4965.2022.067 (In Chinese), we have attached this paper in Data Availability)

”

Supplementary Figure 8. Validation (MAE=mean absolute error; MBE=mean bias error; STD=standard deviation) of cloud top height (CTH) retrieved from Himawari-8 satellite using CALIPSO (Cloud-Aerosol Lidar and Infrared Pathfinder Satellite Observation) product. (Supplementary Figure 8 cited from Reference-3: Min Min, Jun Li, Fu Wang, Zijing Liu, W. Paul Menzel, 2020. Retrieval of cloud top properties from advanced geostationary satellite imager measurements based on machine learning algorithms [J]. Remote Sensing of Environment, 239: 111616, doi: 10.1016/j.rse.2019.111616)

I still find the layout of the paper confusing; with results first and discussion second, followed by methods, I still find myself flipping back and forth between the paper to interpret the results. This is a choice for the editor and the authors to consider, of course.

Response: Thanks for your valuable suggestion. We sincerely apologize for any inconvenience caused to your review due to the format of our manuscript. The current layout of the manuscript may indeed bring many inconveniences to readers, but the layout is determined by the editor. I believe that the editor will provide a reasonable layout for our manuscript. Thank you again for your valuable suggestion.

Reviewer #2 (Remarks to the Author):

Thanks for thoroughly addressing the comments and concerns from the previous round of review. No further suggestions from my end.

Response: Thank you very much for your valuable suggestions. Your suggestions have made our manuscript more reasonable and perfect, and we have also learned a lot during the process of responding to your comments. Thank you again and best wishes.

Reviewer #3 (Remarks to the Author):

I think the authors have made a good revision work. The reviewers' response has cleared all my concerns about the manuscript. I agree with the authors about not providing a tilted plane solar irradiance evaluation, which includes further considerations than the forecast itself. Most PV plants also record the global horizontal irradiance (GHI), and my previous comment referred to this magnitude. I also agree with not providing the discrete-values scatter plots, as they are uninformative. **The non-inclusion of the forecasting skill metric, however, is debatable, as it can be easily constructed from CF persistence. Maybe I am losing something here, but the authors' reply on this issue was not clear to me.**

Response: Thank you very much for your valuable suggestion. Forecasting skill metric is indeed a good parameter for measuring the accuracy of forecasting methods, but its use is based on the premise that different methods predict the same thing under the same conditions. In this study, our predicted result is geostationary satellite L1B radiance data, and then we use cloud mask algorithms to obtain Cloud Fraction over solar PV plants. There is no other study adopted our idea and method. Some previous study

(Marchesoni-Acland et al., 2023) only used a single visible channel of geostationary satellite and a constant model to predict cloudiness (see Q2 of Reviewer-3). While we used predicted six channels (two visible and four infrared) of AHI/H8 to determine cloudy pixels over PV plants. More importantly, our prediction range is very small, only covering an area of $128\text{km} \times 128\text{km}$. In contrast, many other studies (Marchesoni-Acland et al., 2023, Nielsen AH et al. 2021.) have a large prediction range, even covering the entire area of France.

Of course, we understand that it is very difficult for different studies to have exactly the same initial conditions. We try to find similar ones that can be compared with our research, but this is still difficult. We have not found a suitable former study to compare with our method. With the development of neural networks, in our future research, we may adopt more advanced neural network models, so that we can use forecast skills to compare the predictive performance of different models.

Reference:

Marchesoni-Acland F, Herrera A, Mozo F, Camiruaga I, Castro A, Alonso-Suárez R. Deep learning methods for intra-day cloudiness prediction using geostationary satellite images in a solar forecasting framework. *Solar Energy* 262, (2023).

Nielsen AH, Iosifidis A, Karstoft H. IrradianceNet: Spatiotemporal deep learning model for satellite-derived solar irradiance short-term forecasting. *Solar Energy* **228**, 659-669 (2021).

My only comment left is that the reviewers' response frames the article better in the field than the current manuscript, in which there is no mention of previous satellite solar or cloud nowcasting methods based on neural networks. By this, it may be interpreted that this is the only solar satellite nowcasting work based

on satellite images and neural networks, which it is not. In my opinion, the current manuscript's introduction reads as disconnected from the specific state-of-the-art.

Response: Thank you very much for your valuable suggestions. We have updated the last paragraph of the manuscript's introduction a line 96-155 (see below). In this part, we have added some similar previous papers here.

“Cloud cover nowcasting remains a field of interest for forecasting the electricity production of PV plants²⁴. We are committed to developing a daytime hourly intra-day CF prediction algorithm for small areas over PV plants. Based on the recurrent-neural-networks-based (RNNs) long short-term memory (LSTM) algorithm framework, the newly developed PredRNN and PredRNN++ (an extended and latest version of PredRNN)^{25, 26} can well learn to predict long-term future imageries in various spatio-temporal tasks by modeling their spatial and temporal dependencies, including video frame prediction, human motion prediction, etc. Therefore, our primary objective is to develop an innovative and easy-to-promote algorithm or system based on the key framework of the PredRNN++ model. Through this algorithm, the 0 – 4 hour CF at solar PV plants under all weather conditions can be predicted by using sequential Himawari-8/9 geostationary satellite images with high spatio-temporal resolutions²⁷. Compared with the previous study²⁸, it only used a single visible channel of geostationary satellite and a constant model to predict cloudiness. Some former studies directly used surface solar global horizontal irradiance (GHI) as model input to predict GHI values in the next few hours²⁹, achieving the purpose of estimating the power generation of PV plants. Nevertheless, the presence of clouds is still identified as the primary uncertainty in current surface solar GHI forecasts³⁰. In contrast, our investigation only predicts geostationary satellite Level 1B (L1B) radiance data. With the prediction results of satellite L1B radiance data and accurate cloud detection algorithm, this approach is expected to provide reliable and variable CF information for further improving the predictability of current GTI or PV power generation.”